# Robustly overfitting latents for flexible neural image compression

**Yura Perugachi-Diaz**
Vrije Universiteit Amsterdam
y.m.perugachidiaz@vu.nl

**Arwin Gansekoele**
Centrum Wiskunde
& Informatica
awg@cwi.nl

**Sandjai Bhulai**
Vrije Universiteit Amsterdam
s.bhulai@vu.nl

## Abstract

Neural image compression has made a great deal of progress. State-of-the-art models are based on variational autoencoders and are outperforming classical models. Neural compression models learn to encode an image into a quantized latent representation that can be efficiently sent to the decoder, which decodes the quantized latent into a reconstructed image. While these models have proven successful in practice, they lead to sub-optimal results due to imperfect optimization and limitations in the encoder and decoder capacity. Recent work shows how to use stochastic Gumbel annealing (SGA) to refine the latents of pre-trained neural image compression models. We extend this idea by introducing SGA+, which contains three different methods that build upon SGA. We show how our method improves the overall compression performance in terms of the R-D trade-off, compared to its predecessors. Additionally, we show how refinement of the latents with our best-performing method improves the compression performance on both the Tecnick and CLIC dataset. Our method is deployed for a pre-trained hyperprior and for a more flexible model. Further, we give a detailed analysis of our proposed methods and show that they are less sensitive to hyperparameter choices. Finally, we show how each method can be extended to three- instead of two-class rounding.

## 1 Introduction

Image compression allows efficient sending of an image between systems by reducing their size. There are two types of compression: lossless and lossy. Lossless image compression sends images perfectly without losing any quality and can thus be restored in their original format, such as the PNG format. Lossy compression, such as BPG Bellard (2014), JPEG Wallace (1992) or JPEG2000 Skodras et al. (2001), loses some quality of the compressed image. Lossy compression aims to preserve as much of the quality of the reconstructed image as possible, compared to its original format, while allowing a significantly larger reduction in required storage.

Traditional methods Wallace (1992); Skodras et al. (2001), especially lossless methods, can lead to limited compression ratios or degradation in quality. With the rise of deep learning, neural image compression is becoming a popular method Theis et al. (2017); Toderici et al. (2017). In contrast with traditional methods, neural image compression methods have been shown to achieve higher compression ratios and less degradation in image quality Ballé et al. (2018); Minnen et al. (2018); Lee et al. (2019). Additionally, neural compression techniques have shown improvements compared to traditional codecs for other data domains, such as video. Agustsson et al. (2020); Habibian et al. (2019); Lu et al. (2019).

In practice, neural lossy compression methods have proven to be successful and achieve state-of-the-art performance Ballé et al. (2018); Cheng et al. (2020); Minnen et al. (2018); Lee et al. (2019). These models are frequently based on variational autoencoders (VAEs) with an encoder-decoder structure

38th Conference on Neural Information Processing Systems (NeurIPS 2024).

Kingma and Welling (2013). The models are trained to minimize the expected rate-distortion (R-D) cost: $R + \lambda D$. Intuitively, one learns a mapping that encodes an image into a compressible latent representation. The latent representation is sent to a decoder and is decoded into a reconstructed image. The aim is to train the compression model in such way that it finds a latent representation that represents the best trade-off between the length of the bitstream for an image and the quality of the reconstructed image. Even though these models have proven to be successful in practice, they do have limited capacity when it comes to optimization and generalization. For example, the encoder's capacity is limited which makes the latent representation sub-optimal Cremer et al. (2018). Recent work Campos et al. (2019); Guo et al. (2020); Yang et al. (2020) proposes procedures to refine the encoder or latents, which lead to better compression performance. Furthermore, in neural video compression, other work focuses on adapting the encoder Aytekin et al. (2018); Lu et al. (2020) or finetuning a full compression model after training to improve the video compression performance van Rozendaal et al. (2021).

The advantage of refining latents Campos et al. (2019); Yang et al. (2020) is that improved compression results per image are achieved while the model does not need to be modified. Instead, the latent representations for each individual image undergo a refining procedure. This results in a latent representation that obtains an improved bitstream and image quality over its original state from the pre-trained model. As mentioned in Yang et al. (2020), the refining procedure for stochastic rounding with Stochastic Gradient Gumbel Annealing (SGA) considerably improves performance.

In this paper, we introduce SGA+, an extension of SGA that further improves compression performance and is less sensitive to hyperparameter choices. The main contributions are: *(i)* showing how changing the probability space with more natural methods instead of SGA boosts the compression performance, *(ii)* proposing the sigmoid scaled logit (SSL), which can smoothly interpolate between the approximate `atanh`, linear, cosine and round, *(iii)* demonstrating a generalization to rounding to three classes, that contains the two classes as a special case, and *(iv)* showing that SGA+ not only outperforms SGA on a similar pre-trained mean-scale hyperprior model as in Yang et al. (2020), but also achieves an even better performance for the pre-trained models of Cheng et al. (2020). Further, we show how SSL outperforms baselines in an R-D plot on the Kodak dataset, in terms of peak signal-to-noise ratio (PSNR) versus the bits per pixel (BPP) and in terms of true loss curves. Additionally, we show how our method generalizes to the Tecnick and CLIC dataset, followed by qualitative results. We analyze the stability of all functions and show the effect of interpolation between different methods with SSL. Lastly, we analyze a refining procedure at compression time that allows moving along the R-D curve when refining the latents with another $\lambda$ than a pre-trained model is trained on Gao et al. (2022); Xu et al. (2023). The code can be retrieved from: `https://github.com/yperugachidiaz/flexible_neural_image_compression`.

## 2 Preliminaries and related work

In lossy compression, the aim is to find a mapping of image $x$ where the distortion of the reconstructed image $\hat{x}$ is as little as possible compared to the original one while using as little storage as possible. Therefore, training a lossy neural image compression model presents a trade-off between minimizing the length of the bitstream for an image and minimizing the distortion of the reconstructed image Ballé et al. (2017); Lee et al. (2019); Minnen et al. (2018); Theis et al. (2017).

Neural image compression models from Ballé et al. (2017); Cheng et al. (2020); Minnen et al. (2018); Theis et al. (2017), also known as hyperpriors, accomplish this kind of mapping with latent variables. An image $x$ is encoded onto a latent representation $y = g_a(x)$, where $g_a(\cdot)$ is the encoder. Next, $y$ is quantized $Q(y) = \hat{y}$ into a discrete variable that is sent losslessly to the decoder. The reconstructed image is given by: $\hat{x} = g_s(\hat{y})$, where $g_s(\cdot)$ represents the decoder. The rate-distortion objective that needs to be minimized for this specific problem is given by:

$$
\begin{aligned}
\mathcal{L} &= R + \lambda D \\
&= \underbrace{\mathbb{E}_{x \sim p_x} \left[ -\log_2 p_{\hat{y}}(\hat{y}) \right]}_{\text{rate}} + \lambda \underbrace{\mathbb{E}_{x \sim p_x} \left[ d(x, \hat{x}) \right]}_{\text{distortion}},
\end{aligned}
\tag{1}
$$

where $\lambda$ is a Lagrange multiplier determining the rate-distortion trade-off, $R$ is the expected bitstream length to encode $\hat{y}$ and $D$ is the metric to measure the distortion of the reconstructed image $\hat{x}$

compared to the original one $x$. Specifically for the rate, $p_x$ is the (unknown) image distribution and $p_{\hat{y}}$ represents the entropy model that is learned over the data distribution $p_x$. A frequently used distortion measure for $d(x, \hat{x})$, is the mean squared error (MSE) or PSNR.

In practice, the latent variable $y$ often consists of multiple levels in neural compression. Namely, a smaller one named $z$, which is modeled with a relatively simple distribution $p(z)$, and a larger variable, which is modeled by a distribution for which the parameters are predicted with a neural network using $z$, the distribution $p(y|z)$. We typically combine these two variables into a single symbol $y$ for brevity. Furthermore, a frequent method of quantizing $Q(\cdot)$ used to train hyperpriors consists of adding uniform noise to the latent variable.

## 2.1   Latent optimization

Neural image compression models have been trained over a huge set of images to find an optimal encoding. Yet, due to difficulties in optimization or due to constraints on the model capacity, model performance is sub-optimal. To overcome these issues, another type of optimizing compression performance is proposed in Campos et al. (2019); Yang et al. (2020) where they show how to find better compression results by utilizing pre-trained networks and keeping the encoder and decoder fixed but only adapting the latents. In these methods, a latent variable $y$ is iteratively adapted using differentiable operations at test time. The aim is to find a more optimal discrete latent representation $\hat{y}$. Therefore, the following minimization problem needs to be solved for an image $x$:

$$\arg \min_{\hat{y}} \left[ - \log_2 p_{\hat{y}}(\hat{y}) + \lambda d(x, \hat{x}) \right]. \tag{2}$$

This is a powerful method that can fit to a test image $x$ directly without the need to further train an entire compression model.

## 2.2   Stochastic Gumbel Annealing

Campos et al. (2019) proposes to optimize the latents by iteratively adding uniform noise and updating its latents. While this method proves to be effective, there is still a difference between the true rate-distortion loss ($\hat{\mathcal{L}}$) for the method and its discrete representation $\hat{y}$. This difference is also known as the discretization gap. Therefore, Yang et al. (2020) propose the SGA method to optimize latents and show how it obtains a smaller discretization gap. SGA is a soft-to-hard quantization method that quantizes a continuous variable $v$ into the discrete representation for which gradients can be computed. A variable $v$ is quantized as follows. First, a vector $\mathbf{v}_r = (\lfloor v \rfloor, \lceil v \rceil)$ is created that stacks the floor and ceil of the variable, also indicating the rounding direction. Next, the variable $v$ is centered between $(0, 1)$ where for the flooring: $v_L = v - \lfloor v \rfloor$ and ceiling: $v_R = \lceil v \rceil - v$. With a temperature rate $\tau \in (0, 1)$, that is decreasing over time, this variable determines the soft-to-hardness where 1 indicates training with a fully continuous variable $v$ and 0 indicates training while fully rounding variable $v$. To obtain unnormalized log probabilities (logits), the inverse hyperbolic tangent ($\mathrm{atanh}$) function is used as follows:

$$logits = (- \mathrm{atanh}(v_L)/\tau, - \mathrm{atanh}(v_R)/\tau). \tag{3}$$

To obtain probabilities a softmax is used over the $logits$, which gives the probability $p(y)$ which is the chance of $v$ being floored: $p(y = \lfloor v \rfloor)$, or ceiled: $p(y = \lceil v \rceil)$. This is approximated by the Gumbel-softmax distribution. Then, samples are drawn: $\mathbf{y} \sim \text{Gumbel-Softmax}(logits, \tau)$ Jang et al. (2016) and are multiplied and summed with the vector $\mathbf{v}_r$ to obtain the quantized representation: $\hat{v} = \sum_i (v_{r,i} * y_i)$. As SGA aids the discretization gap, this method may not have optimal performance and may not be as robust to changes in its temperature rate $\tau$.

Besides SGA, Yang et al. (2020) propose deterministic annealing Agustsson et al. (2017), which follows almost the same procedure as SGA, but instead of sampling stochastically from the Gumbel Softmax, this method uses a deterministic approach by computing probabilities with the Softmax from the $logits$. In practice, this method has been shown to suffer from unstable optimization behavior.

## 2.3   Other methods

While methods such as SGA aim to optimize the latent variables for neural image compression at inference time, other approaches have been explored in recent research. Guo et al. (2021) proposed a soft-then-hard strategy alongside a learned scaling factor for the uniform noise to achieve better

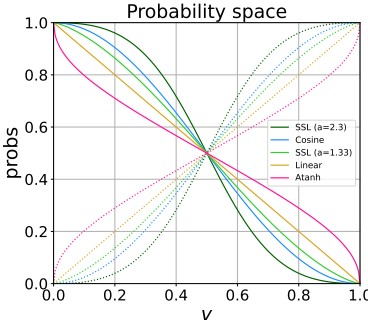 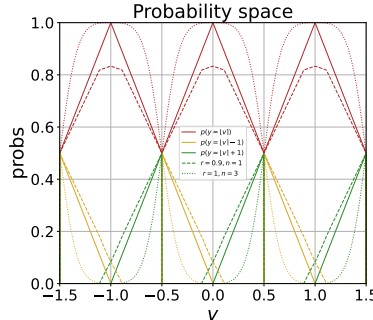

(a) Probability space for several SGA+ methods, along with $\mathrm{atanh}$. Solid lines denote the probability of flooring $\lfloor v \rfloor$ and dotted lines the probability of ceiling $\lceil v \rceil$.

(b) Three-class rounding for the extended version of linear. Solid lines denote two-class rounding, dashed lines denote three-class rounding and dotted lines indicate the smoothness.

Figure 1: Probability space for (a) Two-class rounding (b) Three-class rounding

compression and a smoother latent. These methods are used to fine-tune network parameters but not the latents directly. Zhu et al. (2022) proposed using Swin-transformer-based coding instead of ConvNet-based coding. They showed that these transforms can achieve better compression with fewer parameters and shorter decoding times. van Rozendaal et al. (2021) proposed to also fine-tune the decoder alongside the latent for video compression. While accommodating the additional cost of saving the model update, they demonstrated a gain of $\sim 1dB$. Zhang et al. (2021) and Dupont et al. (2021) proposed using implicit neural representations for video and image compression, respectively. He et al. (2022) proposed an improved context model (SCCTX) and a change to the main transform (ELIC) that achieve strong compression results together. El-Nouby et al. (2023) revisited vector quantization for neural image compression and demonstrated it performs on par with hyperprior-based methods. Li et al. (2020) proposed a method to incorporate trellis-coded quantization in neural codecs. While these approaches change the training process, our work differs in that we only consider the inference process. Balcilar et al. (2023) proposes latent shift, a method that can further optimize latents using the correlation between the gradient of the reconstruction error and the gradient of the entropy.

## 3 Methods

As literature has shown, refining the latents of pre-trained compression models with SGA leads to improved compression performance Yang et al. (2020). In this section, we extend SGA by introducing SGA+ containing three other methods for the computation of the unnormalized log probabilities ($logits$) to overcome issues from its predecessor. We show how these methods behave in probability space. Furthermore, we show how the methods can be extended to three-class rounding.

### 3.1 Two-class rounding

Recall from SGA that a variable $v$ is quantized to indicate the rounding direction to two classes and is centered between (0,1). Computation of the unnormalized log probabilities is obtained with $\mathrm{atanh}$ from Equation (3). Recall, that in general the probabilities are given by a softmax over the *logits* with a function of choice. As an example, for SGA the logits are computed with $\mathrm{atanh}$. The corresponding probabilities for rounding down is then equal to: $\frac{e^{\mathrm{atanh}(v_L)}}{e^{\mathrm{atanh}(v_L)} + e^{\mathrm{atanh}(v_R)}}$. Then looking at the probability space from this function, see Figure 1a, the $\mathrm{atanh}$ function can lead to sub-optimal performance when used to determine rounding probabilities. The problem is that gradients tend to infinity when the function approaches the limits of 0 and 1, see Appendix A for the proof that gradients at 0 tend to $\infty$. This is not ideal, as these limits are usually achieved when the discretization gap is minimal. In addition, the gradients may become larger towards the end of optimization. Further analyzing the probability space, we find that there are a lot of possibilities in choosing probabilities for rounding to two classes. However, there are some constraints: the probabilities need to be monotonic functions, and the probabilities for rounding down (flooring) and up (ceiling) need to sum up to one. Therefore, we introduce SGA+ and propose three methods that satisfy the above constraints and can be used to

overcome the sub-optimality that the atanh function suffers from. We opted for these three as they each have their own interesting characteristics. However, there are many other functions that are also valid and would behave similarly to these three.

We will denote the probability that $v$ is rounded down by:

$$p(y = \lfloor v \rfloor), \tag{4}$$

where $y$ represents the random variable whose outcome can be either rounded down or up. The probability that $v$ is rounded up is conversely: $p(y = \lceil v \rceil) = 1 - p(y = \lfloor v \rfloor)$.

**Linear probabilities**  To prevent gradient saturation or vanishing gradients completely, the most natural case would be to model a probability that linearly increases or decreases and has a gradient of one everywhere. Therefore, we define the linear:

$$p(y = \lfloor v \rfloor) = 1 - (v - \lfloor v \rfloor). \tag{5}$$

It is easy to see that: $p(y = \lceil v \rceil) = v - \lfloor v \rfloor$. In Figure 1a, the linear probability is shown.

**Cosine probabilities**  As can be seen in Figure 1a, the atanh tends to have gradients that go to infinity for $v$ close to the corners. Subsequently, a method that has low gradients in that area is by modeling the cosine probability as follows:

$$p(y = \lfloor v \rfloor) = \cos^2 \left( \frac{(v - \lfloor v \rfloor)\pi}{2} \right). \tag{6}$$

This method aids the compression performance compared to the atanh since there is less probability of overshooting the rounding value.

**Sigmoid scaled logit**  There are a lot of possibilities in choosing probabilities for two-class rounding. We introduced two probabilities that overcome sub-optimality issues from atanh: the linear probability from Equation (5), which has equal gradients everywhere, and cosine from Equation (6), that has little gradients at the corners. Besides these two functions, the optimal probability might follow a different function from the ones already mentioned. Therefore, we introduce the sigmoid scaled logit (SSL), which can interpolate between different probabilities with its hyperparameter $a$ and is defined as follows:

$$p(y = \lfloor v \rfloor) = \sigma(-a\sigma^{-1}(v - \lfloor v \rfloor)), \tag{7}$$

where $a$ is the factor determining the shape of the function. SSL is exactly the linear for $a = 1$. For $a = 1.6$ and $a = 0.65$ SSL roughly resembles the cosine and atanh. For $a \to \infty$ the function tends to shape to (reversed) rounding.

Note that the main reason behind the linear version is the fact that it is the only function with constant gradients which is also the most robust choice, the cosine version is approximately mirrored across the diagonal of the $-\operatorname{atanh}(x)$ which shows that it is more stable compared to the $-\operatorname{atanh}(x)$, and the reason behind the SSL is that it is a function that can interpolate between all possible functions and can be tuned to find the best possible performance when necessary.

## 3.2 Three-class rounding

As described in the previous section, the values for $v$ can either be floored or ceiled. However, there are cases where it may help to round to an integer further away. Therefore, we introduce three-class rounding and show three extensions that build on top of the linear probability Equation (5), cosine probability Equation (6), and SSL from Equation (7).

The probability that $v$ is rounded is denoted by: $p(y = \lfloor v \rfloor) \propto f_{3c}(w|r, n)$, where $w = v - \lfloor v \rceil$ is centered around zero. Further, we define the probability that $v$ is rounded $+1$ and rounded $-1$ is respectively given by: $p(y = \lfloor v \rceil - 1) \propto f_{3c}(w - 1|r, n)$ and $p(y = \lfloor v \rceil + 1) \propto f_{3c}(w + 1|r, n)$. Recall, that $v_L, v_R \in [0, 1]$, whereas $w \in [-0.5, 0.5]$. Defining $w$ like this is more helpful for the 3-class since it has a center class. The general notation for the three-class functions is given by:

$$f_{3c}(w|r, n) = f(\operatorname{clip}(w \cdot r))^n, \tag{8}$$

where $\operatorname{clip}(\cdot)$ clips the value at 0 and 1, $r$ is the factor determining the height and steepness of the function and power $n$ controls the peakedness of the function. Note that $n$ can be fused with temperature $\tau$ together, to scale the function. This only accounts for the computation of the logits and not to modify the Gumbel temperature, therefore, $\tau$ still needs a separate definition.

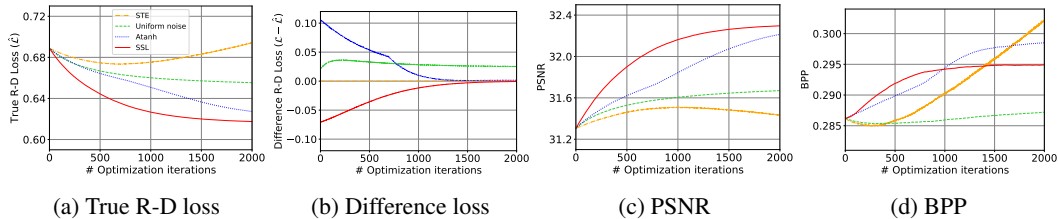

| (a) True R-D loss | (b) Difference loss | (c) PSNR | (d) BPP |

Figure 2: Performance plots of (a) True R-D Loss (b) Difference in loss (c) PSNR (d) BPP.

**Extended linear**   Recall that the linear probability can now be extended to three-class rounding as follows:

$$f_{linear}(w) = |w|. \tag{9}$$

A special case is $f_{3c,linear}(w|r = 1, n = 1)$, where the function is equivalent to the linear of the two-class rounding from Equation (5). For $r < 1$ this function rounds to three classes, and for $n \neq 1$ this function is not linear anymore.

In Figure 1b, three-class rounding for the extension of Equation (5) can be found. As can be seen, solid lines denote the special case of two-class rounding with $r = 1$ and $n = 1$, dashed lines denote three-class rounding with $r = 0.9$ and $n = 1$ and dotted lines denote the two-class rounding with $r = 1$ and $n = 3$, which shows a less peaked function. For an example of two- versus three-class rounding, consider the case where we have variable $v = -0.95$. For two-class rounding there is only the chance of rounding to $-1$ with $p(y = \lfloor v \rceil)$ (red solid line), a chance to round to $0$ with $p(y = \lfloor v \rceil + 1)$ (green solid line) and zero chance to round to $-2$ with $p(y = \lfloor v \rceil - 1)$ (yellow solid line). For three-class rounding, with $r = 0.9$ and $n = 1$, when $v = -0.95$ we find a high chance to round to $-1$ with $p(y = \lfloor v \rceil)$ (red dashed line) and a small chance to round to $0$ with $p(y = \lfloor v \rceil + 1)$ (green dashed line) and a tiny chance to round to $-2$ with $p(y = \lfloor v \rceil - 1)$ (yellow dashed line).

**Extended cosine**   Similarly, we can transform the cosine probability from Equation (6) to three-class rounding:

$$f_{cosine}(w) = \cos\left(\frac{|w|\pi}{2}\right). \tag{10}$$

When $f_{3c,cosine}(w|r = 1, n = 2)$, this function exactly resembles the cosine for two-class rounding, and for $r < 1$ this function rounds to three classes.

**Extended SSL**   Additionally, SSL from Equation (7) can be transformed to three-class rounding as follows:

$$f_{SSL}(w) = \sigma\left(-a\sigma^{-1}(|w|)\right), \tag{11}$$

where $a$ is the factor determining the shape of the function. When $f_{3c,SSL}(w|r = 1, n = 1)$, this function exactly resembles the two-class rounding case, and for $r < 1$, the function rounds to three classes. Recall that this function is capable of exactly resembling the linear function and approximates the cosine from two-class rounding for $a = 1$ and $a = 1.6$, respectively.

## 4   Experiments

In this section, we show the performance of our best-performing method in an R-D plot and compare it to the baselines. Further, we evaluate and compare the methods with the true R-D loss performance $(\hat{\mathcal{L}})$, the difference between the method loss and true loss $(\mathcal{L} - \hat{\mathcal{L}})$, and corresponding PSNR and BPP plot that respectively expresses the image quality, and cost over $t$ training steps. Finally, we show how our best-performing method performs on the Tecnick and CLIC dataset and show qualitative results.

Following Yang et al. (2020), we run all experiments with temperature schedule $\tau(t) = \min(\exp\{-ct\}, \tau_{max})$, where $c$ is the temperature rate determining how fast temperature $\tau$ is decreasing over time, $t$ is the number of train steps for the refinement of the latents and $\tau_{max} \in (0, 1)$ determines how soft the latents start the refining procedure. Additionally, we refine the latents for $t = 2000$ train iterations, unless specified otherwise. See Section 4.2 for the hyperparameter settings.

**Implementation details** We use two pre-trained hyperprior models to test SGA+, for both models we use the package from CompressAI Bégaint et al. (2020). The first model is similar to the one trained in Yang et al. (2020). Note, that the refining procedure needs the same storage and memory as theirs. In Appendix C, implementation details and results of this model can be found. All experiments in this section are run with a more recent hyperprior-based model which is based on the architecture of Cheng et al. (2020). Additionally, this model reduces the R-D loss for $\mathtt{atanh}$ on average by $7.6\%$ and for SSL by $8.6\%$, compared to the other model. The model weights can be retrieved from CompressAI Bégaint et al. (2020). The models were trained with $\lambda = \{0.0016, 0.0032, 0.0075, 0.015, 0.03, 0.045\}$. The channel size is set to $N = 128$ for the models with $\lambda = \{0.0016, 0.0032, 0.0075\}$, refinement of the latents on Kodak with these models take approximately 21 minutes. For the remaining $\lambda$'s channel size is set to $N = 192$ and the refining procedure takes approximately 35 minutes. We perform our experiments on a single NVIDIA A100 GPU.

**Baseline methods** We compare our methods against the methods that already exist in the literature. The **Straight-Through Estimator** (STE) is a method to round up or down to the nearest integer with rounding bound set to a half. This rounding is noted as $\lfloor \cdot \rceil$. The derivative of STE for the backward pass is equal to 1 Bengio et al. (2013); Van Den Oord et al. (2017); Yin et al. (2019). The **Uniform Noise** quantization method adds uniform noise from $u \sim U(-\frac{1}{2}, \frac{1}{2})$ to latent variable $y$. Thus: $\hat{y} = y + u$. In this manner $\hat{y}$ becomes differentiable Ballé et al. (2017). As discussed in Section 2.2, we compare against **Stochastic Gumbel Annealing**, which is a soft-to-hard quantization method that quantizes a continuous variable $v$ into a discrete representation for which gradients can be computed.

## 4.1 Overall performance

Figure 3a shows the rate-distortion curve, using image quality metric PSNR versus BPP, of the base model and for refinement of the latents on the Kodak dataset with method: STE, uniform noise, $\mathtt{atanh}$ and SSL. We clearly see how SSL outperforms all other methods. In Appendix B.1a, the results after $t = 500$ iterations are shown, where the performance is more pronounced.

To show how each of the methods behave, we take a closer look at the performance plots shown in Figure 2. The refinement results are obtained from the pre-trained model, trained with $\lambda = 0.0075$. The true R-D loss in Figure 2a shows that STE performs worse and has trouble converging, which is reflected in the R-D curve. Uniform noise quickly converges compared to $\mathtt{atanh}$ and SSL. We find that SSL outperforms all other methods, including $\mathtt{atanh}$ in terms of the lowest true R-D loss at all steps. Looking at the difference between the method loss and true loss Figure 2b, we find that both SSL and $\mathtt{atanh}$ converge to 0. Yet, the initial loss difference is smaller and smoother for SSL, compared to $\mathtt{atanh}$. Additionally, uniform noise shows a big difference between the method and true loss, indicating that adding uniform noise overestimates its method loss compared to the true loss.

**Tecnick and CLIC** To test how our method performs on other datasets, we use the Tecnick Asuni and Giachetti (2014) and CLIC dataset. We run baselines $\mathtt{atanh}$ and the base model and compare against SSL with $a = 2.3$. Figure 3b shows the corresponding R-D curves on the Tecnick dataset. Note that Appendix B.3 contains the results of the CLIC dataset which shows similar behavior as on Tecnick. We find that both refining methods improve the compression performance in terms of the R-D trade-off. Additionally, our proposed method outperforms $\mathtt{atanh}$ and shows how it improves performance significantly at all points. The R-D plot after $t = 500$ iterations for both datasets can be found in Appendix B.1. Note, that these results show even more pronounced performance difference.

**BD-Rate and BD-PSNR** To evaluate the overall improvements of our method, we computed the BD-Rate and BD-PSNR Bjontegaard for Kodak, Tecnick and CLIC in Table 1. We observe that across the board SSL achieves an improvement in both BD-PSNR and BD-Rate. After 500 steps, SSL achieves almost double the BD-Rate reduction as atanh. The difference between the two becomes smaller at 2000 steps. This underlines the faster convergence behavior of the SSL method.

## 4.2 Qualitative results

In Figure 4, we demonstrate the visual effect of our approach. We compare the original image with the compressed image from base model with $\lambda = 0.0016$, refinement method $\mathtt{atanh}$, and SSL. For

Table 1: Pairwise Comparison between atanh and SSL of BD-PSNR and BD-Rate.

| | BD-PSNR (dB) | | | | | | BD-Rate (%) | | | | | |
| | 500 steps | | | 2000 steps | | | 500 steps | | | 2000 steps | | |
| | Kodak | Tecnick | CLIC | Kodak | Tecnick | CLIC | Kodak | Tecnick | CLIC | Kodak | Tecnick | CLIC |
|---|---|---|---|---|---|---|---|---|---|---|---|---|
| Base vs SSL | 0.50 | 0.57 | 0.56 | 0.82 | 0.95 | 0.89 | -10.30 | -11.60 | -13.18 | -16.23 | -18.77 | -20.11 |
| Base vs Atanh | 0.26 | 0.28 | 0.31 | 0.69 | 0.79 | 0.78 | -5.52 | -5.91 | -7.37 | -13.82 | -15.93 | -17.70 |
| Atanh vs SSL | 0.24 | 0.28 | 0.26 | 0.14 | 0.16 | 0.11 | -5.04 | -5.97 | -6.20 | -2.86 | -3.34 | -2.76 |

the image compressed by SSL, we observe that there are less artifacts visible in the overall image. For instance, looking at the window we see more texture compared to the base model and atanh method.

**Hyperparameter settings**    Refinement of the latents with pre-trained models, similar to the one trained in Yang et al. (2020), use the same optimal learning rate of $0.005$ for each method. Refinement of the latents with the models of Cheng et al. (2020) use a $10$ times lower learning rate of $0.0005$. Following Yang et al. (2020), we use the settings for atanh with temperature rate $\tau_{max} = 0.5$, and for STE we use the smaller learning rate of $0.0001$, yet STE still has trouble converging. Note that, we tuned STE just as the other baselines. However, the STE method is the only method that has a lot of trouble converging. Even with smaller learning rates, the method performed poorly. The instability of training is not only observed by us, but is also observed in Yang et al. (2020); Yin et al. (2019). For SGA+, we use optimal convergence settings, which are a fixed learning rate of $0.0005$, and $\tau_{max} = 1$. Experimentally, we find approximately best performance for SLL with $a = 2.3$.

## 5 Analysis and Societal Impact

In this section we perform additional experiments to get a better understanding of SGA+. An in-depth analysis shows the stability of each proposed method, followed by an experiment that expresses changes in the true R-D loss performance when one interpolates between functions. Further, we evaluate three-class rounding for each of our methods. Finally, we show how SGA+ for semi-multi-rate behavior improves the performance of its predecessor and we discuss the societal impact. Note, the results of the additional experiments are obtained from the model trained with $\lambda = 0.0075$.

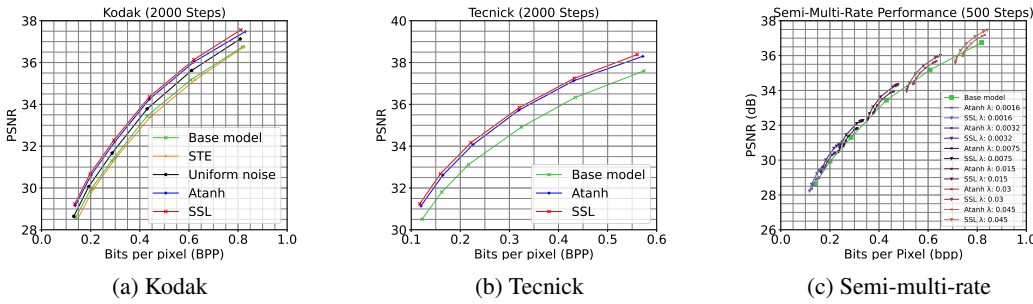

| (a) Kodak | (b) Tecnick | (c) Semi-multi-rate |

Figure 3: R-D performance for SSL on (a) Kodak with the baselines, (b) Tecnick with the base model and atanh and (c) Kodak for semi-multi-rate behavior with atanh. Best viewed electronically.

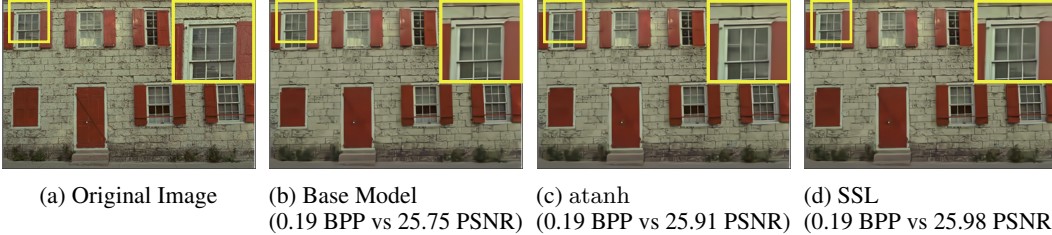

| (a) Original Image | (b) Base Model | (c) atanh | (d) SSL |
| | (0.19 BPP vs 25.75 PSNR) | (0.19 BPP vs 25.91 PSNR) | (0.19 BPP vs 25.98 PSNR) |

Figure 4: Qualitative comparison of a Kodak image from pre-trained model trained with $\lambda = 0.0016$. Best viewed electronically.

**Temperature sensitivity**  Table 2 represents the stability of $\mathtt{atanh}$ and the SGA+ methods, expressed in true R-D loss, for different $\tau_{max}$ settings for the temperature schedule. As can be seen, the most optimal setting is around $\tau_{max} = 1$ for each of the methods. In the table we find that the linear function is least sensitive to changes in $\tau_{max}$. To further examine the stability of the linear function compared to $\mathtt{atanh}$, we subtract the best $\tau_{max}$,

Table 2: True R-D loss for different $\tau_{max}$ settings of: $\mathtt{atanh}(v)$, linear, cosine and SSL with $a = 2.3$. The lowest R-D loss per column is marked with: $\downarrow$. Note that the function containing $\mathtt{atanh}$ is unnormalized.

| Function \ $\tau_{max}$ | 0.2 | 0.4 | 0.6 | 0.8 | 1.0 |
|---|---|---|---|---|---|
| $\exp \mathtt{atanh}(v)$ | 0.6301 | 0.6273 | 0.6267 | 0.6260 | 0.6259 |
| $1 - v$ (linear) | 0.6291 $\downarrow$ | 0.6229 $\downarrow$ | 0.6225 | 0.6222 | 0.6220 |
| $\cos^2(\frac{v\pi}{2})$ | 0.6307 | 0.6233 | 0.6194 $\downarrow$ | 0.6186 | 0.6187 |
| $\sigma(-a\sigma^{-1}(v))$ | 0.6341 | 0.6233 | 0.6196 | 0.6181 $\downarrow$ | **0.6175** $\downarrow$ |
| $\exp \mathtt{atanh}(v)$ | 0.0010 | 0.0044 | 0.0073 | 0.0079 | 0.0084 |
| $1 - v$ (linear) | 0 | 0 | 0.0031 | 0.0041 | 0.0045 |

column-wise, from the linear and $\mathtt{atanh}$ of that column. Also taking into account the sensitivity results of the one of Yang et al. (2020) in Appendix C.2 we find in general, that overall performance varies little compared to the best $\tau_{max}$ settings of the other methods and has reasonable performance. While SSL has the largest drop in performance when reducing $\tau_{max}$, it achieves the highest performance overall for higher values of $\tau_{max}$. If there is no budget to tune the hyperparameter of SGA+, the linear version is the most robust choice. Further, we evaluated the necessity of tuning both the latents and hyper-latents. When only optimizing the latents with the linear approach for $\tau_{max} = 1$, we found a loss of 0.6234. This is a difference of 0.0012, which implies that optimizing the hyper-latent aids the final loss.

**Interpolation**  Table 4 represents the interpolation between different functions, expressed in true R-D loss. In Appendix B.2 the corresponding loss plots can be found. Values for $a < 1$ indicate methods that tend to have larger gradients for $v$ close to the corners, while high values of $a$ represent a method that tends to a (reversed) step function. The smallest $a = 0.01$ diverges and results in a large loss value compared to the rest. For $a = 2.3$ we find a lowest loss of 0.6175 and for $a = 5$ we find fastest convergence compared to the rest. Comparing these model results with the model results with the one of Yang et al. (2020), see Appendix C.2, we find that the Cheng et al. (2020) model obtains more stable curves.

Table 3: True R-D loss of two- versus three-class rounding for SGA+ with the extended version of the linear, cosine, and SSL method at iteration 2000 and in brackets after 500 iterations.

| Function \ Rounding | Two | Three |
|---|---|---|
| $f_{3c,\text{linear}}(w\|r = 0.98, n = 2.5)$ | 0.6220 (0.6594) | 0.6175 (0.6435) |
| $f_{3c,\text{cosine}}(w\|r = 0.98, n = 3)$ | 0.6187 (0.6516) | 0.6175 (0.6449) |
| $f_{3c,\text{sigmoidlogit}}(w\|r = 0.93, n = 2.5)$ | 0.6175 (0.6445) | 0.6203 (0.6360) |

**Three-class rounding**  In Table 3, the true R-D loss for two versus three-class rounding can be found at iteration $t = 2000$ and in brackets $t = 500$ iterations. For each method, we performed a grid search over the hyperparameters $r$ and $n$. As can be seen in the table, most impact is made with the extended version of the linear of SGA+, in terms of the difference between the two versus three-class rounding at iteration $t = 2000$ with loss difference 0.0045 and $t = 500$ with 0.0159 difference. In Appendix B.3 a loss plot of the two- versus three- class rounding for the extended linear method can be found. Concluding, the three-class converges faster. For the extended cosine version, there is a smaller difference and for SSL we find that the three-class extension only boosts performance when run for $t = 500$. In Appendix C.2, the three-class experiments for the pre-trained model similar to Yang et al. (2020) can be found. We find similar behavior as for the model of Cheng et al. (2020),

Table 4: True R-D loss results for the interpolation between different functions by changing $a$ of the SSL.

| a | R-D Loss |
|---|---|
| 0.01 | 0.7323 |
| 0.3 | 0.6352 |
| 0.65 (approx atanh) | 0.6260 |
| 0.8 | 0.6241 |
| 1 (linear) | 0.6220 |
| 1.33 | 0.6199 |
| 1.6 (approx cosine) | 0.6186 |
| 2.3 | 0.6175 $\downarrow$ |
| 5 | 0.6209 |

whereas with the linear version most impact is made, followed by the cosine and lastly SSL. The phenomenon that three-class rounding only improves performance for $t = 500$ iterations for SSL may be due to the fact that SSL is already close to optimal. Additionally, we ran an extra experiment to asses what percentage of the latents are assigned to the 3-classes. This is run with the best settings for the linear version $f_{3c,\text{linear}}(w\|r = 0.98, n = 2.5)$. At the first iteration, the probability is distributed as follows: $p(y = \lfloor v \rceil) = 0.9329$, for $p(y = \lfloor v \rceil - 1) = 0.0312$, and $p(y = \lfloor v \rceil + 1) = 0.0359$. This indicates that the class probabilities are approximately 3.12 for class $-1$ and 3.6 for class $+1$. This is a lot when taking into account that many samples are taken for a large dimensional latent. In

conclusion, three-class rounding may be attractive under a constraint optimization budget, possibly because it is easier to jump between classes. Additionally, for the extended linear three-class rounding also results in faster convergence.

**Semi-multi-rate behavior**  An interesting observation is that one does not need to use the same $\lambda$ during refinement of the latents, as used during training. This is also mentioned in Gao et al. (2022) for image and Xu et al. (2023) for video compression. As a consequence of this approach, we can optimize to a neighborhood of the R-D curve without the need to train a new model from scratch. We experimented and analyze the results with methods: atanh and SSL. Figure 3c shows the performance after $t = 500$ iterations for the model of Cheng et al. (2020). We find that SSL is moving further along the R-D curve compared to atanh. Note the refinement does not span the entire curve and that the performance comes closer together for running the methods longer, see Appendix B.4. For future work it would be interesting how SGA+ compares to the methods mentioned in Gao et al. (2022); Xu et al. (2023), since SSL outperforms atanh.

**Societal impact**  The improvement of neural compression techniques is important in our data-driven society, as it allows quicker development of better codecs. Better codecs reduce storage and computing needs, thus lowering costs. However, training these codes requires significant computational resources, which harms the environment through power consumption and the need for raw materials.

## 6  Conclusion and Limitations

In this paper we proposed SGA+, a more effective extension for refinement of the latents, which aids the compression performance for pre-trained neural image compression models. We showed how SGA+ has improved properties over SGA and we introduced SSL that can approximately interpolate between all of the proposed methods. Further, we showed how our best-performing method SSL outperforms the baselines in terms of the R-D trade-off and how it also outperforms the baselines on the Tecnick and CLIC dataset. Exploration of SGA+ showed how it is more stable under varying conditions. Additionally, we gave a general notation and demonstrated how the extension to three-class rounding improves the convergence of the SGA+ methods. Lastly, we showed how SGA+ improves the semi-multi-rate behavior over SGA. In conclusion, especially when a limited computational budget is available, SGA+ offers the option to improve the compression performance without the need to re-train an entire network and can be used as a drop-in replacement for SGA.

Besides being effective, SGA+ also comes with some limitations. Firstly, we run each method for 2000 iterations per image. In practice this is extremely long and time consuming. We find that running the methods for 500 iterations already has more impact on the performance and we would recommend doing this, especially when a limited computational budget is available. Future work may focus on reducing the number of iterations and maintaining improved performance. Note, higher values for $a$ flatten out quickly, but they achieve much better gains with low-step budgets. Further, the best results are obtained while tuning the hyperparameter of SSL and for each of our tested models this lead to different settings. Note, that the experiments showed that the linear version of SGA+ is least sensitive to hyperparameter changes and we would recommend using this version when there is no room for tuning. Additionally, although three-class rounding improves the compression performance in general, it comes with the cost of fine-tuning extra hyperparameters. Finally, it applies for each method that as the temperature rate has reached a stable setting, the performance will be less pronounced, the longer you train, but in return requires extra computation time at inference.

## Acknowledgments

This work was carried out on the Dutch national e-infrastructure with the support of SURF Cooperative.

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

# A Proof

In this appendix we will proof that the normalization from the (Gumbel) softmax causes infinite gradients at 0.

Recall that the probability is given by a 2-class softmax is defined by:

$$K(v) = \frac{e^{f(v)}}{e^{f(v)} + e^{g(v)}},$$

where $f(v) = -\text{atanh}(v)$ and $g(v) = -\text{atanh}(1 - v)$. We will study the softmax for the first class 0, since the softmax is symmetric this also holds for the second class. The problem is that the gradients of the function $K(v)$ will tend to $\infty$ for both $v \to 1$ but also for $v \to 0$. Here we show that the gradients also tend $\infty$ for $v \to 0$, via the normalization with the term $g(v)$. First, take the derivative to $v$:

$$\frac{dK(v)}{dv} = \frac{dK(v)}{df(v)} \cdot \frac{df(v)}{dv} + \frac{dK(v)}{dg(v)} \cdot \frac{dg(v)}{dv},$$

where $\frac{dK(v)}{df(v)} = K(v)(1 - K(v))$ and $\frac{dK(v)}{dg(v)} = -K(v)\frac{e^{g(v)}}{e^{f(v)}+e^{g(v)}}$. Recall that $\frac{d\text{atanh}(v)}{dv} = \frac{1}{1-v^2}$ therefore $\frac{df(v)}{dv} = -\frac{1}{1-v^2}$ and $\frac{dg(v)}{dv} = \frac{1}{1-(1-v)^2}$. Plugging this in and computing $\frac{dK(v)}{dg(v)}$ gives us:

$$\frac{dK(v)}{dv} = K(v)(1 - K(v))\left(-\frac{1}{1 - v^2}\right) - K(v) \cdot \frac{e^{g(v)}}{e^{f(v)} + e^{g(v)}} \cdot \frac{1}{1 - (1 - v)^2}.$$

Taking the limit to 0, (recall that $\lim_{v \to 0} K(v) = 1, \lim_{v \to 0} e^{f(v)} = 1$ and $\lim_{v \to 0} e^{g(v)} = 0$) allows the following simplifications:

$$\lim_{v \to 0} 0 \cdot \left(-\frac{1}{1 - 0^2}\right) - 1 \cdot \frac{e^{g(v)}}{1 + 0} \cdot \frac{1}{1 - (1 - v)^2}$$

For simplicity we substitute $q = 1 - v$ (when $q \to 1$, then $v \to 0$) which will result in the following:

$$\lim_{v \to 0} -e^{-\text{atanh}(1-v)} \cdot \frac{1}{1 - (1 - v)^2} = \lim_{q \to 1} -e^{-\text{atanh}(q)} \cdot \frac{1}{1 - q^2},$$

Recall, $-\text{atanh}(q) = -\frac{1}{2} \ln \frac{1+q}{1-q}$, so $e^{-\text{atanh}(q)} = 1/\sqrt{\frac{1+q}{1-q}}$ thus:

$$-\lim_{q \to 1} \sqrt{\frac{1 - q}{1 + 1}} \cdot \frac{1}{1 - q^2} = -\lim_{q \to 1} \sqrt{\frac{1}{2} \frac{(1 - q)}{(1 - q^2)^2}}$$

Since $\frac{1}{2}$ is a constant and $\lim_{x \to \infty} \sqrt{x} = \infty$ the final step is to simplify and solve:

$$-\lim_{q \to 1} \sqrt{\frac{(1 - q)}{(1 - q^2)^2}} = -\lim_{q \to 1} \sqrt{\frac{-1}{(q - 1)(q + 1)^2}} = -\infty.$$

This concludes the proof that the gradients tend to $-\infty$ for $v \to 0$.

# B  Additional results

In this appendix, additional experimental results for refinement of the latents with the pre-trained models of Cheng et al. (2020) can be found.

**Difference across runs**  Although we do not report the standard deviation for every run, the difference across runs is very small. For the model of Cheng et al. (2020) with $\lambda = 0.0032$, running five refinement procedures for 2000 iterations, results in a mean of $0.3969$ and a standard deviation of $2.41 \cdot 10^{-5}$.

## B.1  Additional overall performance

Figure B.1 show the R-D curve for the Kodak dataset. We observe that $\mathtt{atanh}$ is similar to uniform noise at $t = 500$ iterations, while SSL manages to achieve better R-D gains. After $t = 2000$ iterations SLL achieves the best R-D trade-off, but the gain is a bit smaller compared to $\mathtt{atanh}$ at $t = 500$ iterations.

**Tecnick and CLIC**  Figure B.2 and Figure B.3 show the R-D results for respectively, Tecnick and CLIC at $t = \{500, 2000\}$ iterations. We observe that SSL achieves best performance at $t = 2000$, compared to $t = 500$ iterations. However, running the method for 2000 iterations is in practice very long. Looking at the results at $t = 500$ iterations we see that there is already great improvement of performance for SSL, compared to the base model for both Tecnick and CLIC dataset.

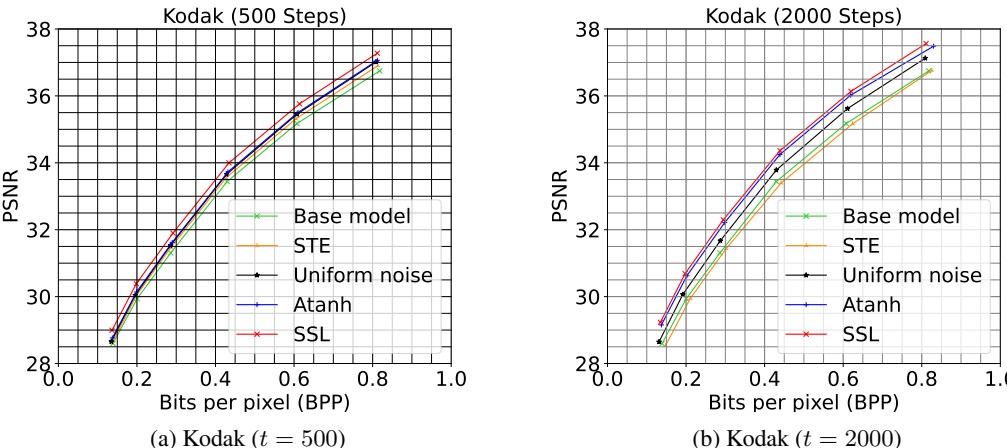

(a) Kodak ($t = 500$)    (b) Kodak ($t = 2000$)

Figure B.1: Comparison of $\mathtt{atanh}$ and SSL on the Kodak dataset for $t = \{500, 2000\}$ iterations.

## B.2  Interpolation

In Figure B.4a the true loss, difference in loss, BPP and PSNR curves can be found. As can be seen for $a = 0.01$, the function diverges and $a = 0.3$ does not seem to reach stable behavior, both resulting in large loss values. For $a \geq 1$, the difference in losses start close to zero (see Figure C.7b). SSL with $a = 5$ results in the fastest convergence and quickly finds a stable point but ends at a higher loss than most methods.

## B.3  Two- versus three- class rounding

Besides an improved RD performance for the three-class rounding, we also found that three-class rounding leads to faster convergence. In Figure B.5 a true loss plot over the iterations for the linear method, can be found. As one can see, the three-class converges faster and especially on 500 iterations, boosts performance which makes it attractive under a constraint budget.

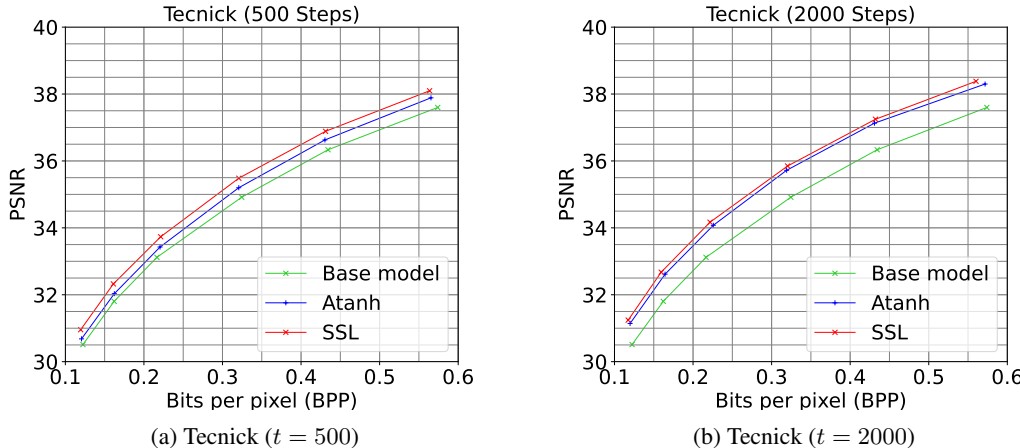

(a) Tecnick ($t = 500$)   (b) Tecnick ($t = 2000$)

Figure B.2: Comparison of $\mathtt{atanh}$ and SSL on the Tecnick dataset for $t = \{500, 2000\}$ iterations.

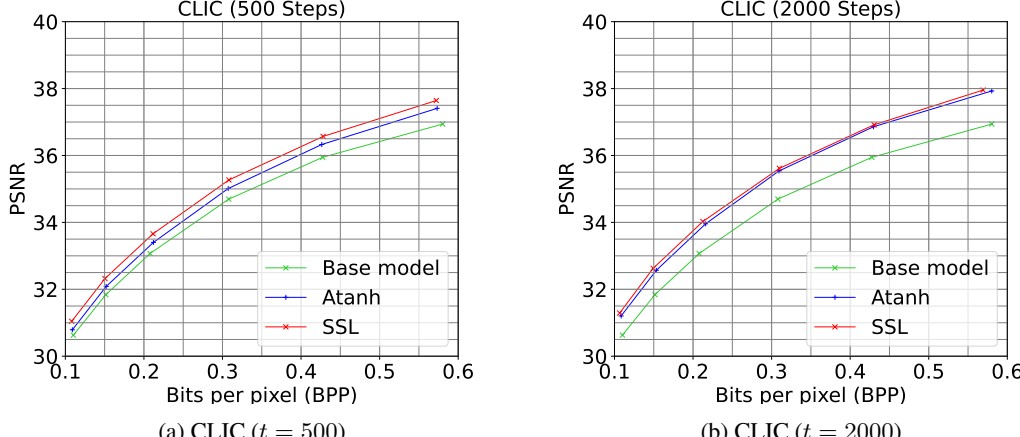

(a) CLIC ($t = 500$)   (b) CLIC ($t = 2000$)

Figure B.3: Comparison of $\mathtt{atanh}$ and SSL on the CLIC dataset for $t = \{500, 2000\}$ iterations.

## B.4   Semi-multi-rate behavior

In Figure B.6, we have plotted the R-D curve of the base model (lime green line) and its corresponding R-D curves, obtained when refining the latents with the proposed $\lambda$'s. For each model trained using $\lambda \in \{0.0016, 0.0032, 0.0075, 0.015, 0.03, 0.045\}$, we run $\mathtt{atanh}$ and SSL with $a = 2.3$ for $t = 2000$ iterations for all $\lambda \in \{0.0004, 0.0008, 0.0016, 0.0032, 0.0075, 0.015, 0.03, 0.045, 0.06, 0.09\}$. We depicted the base curve alongside the curves for each base model. We observe that the improved performance by SSL is especially noticeable at $t = 500$ iterations in Figure B.6a.

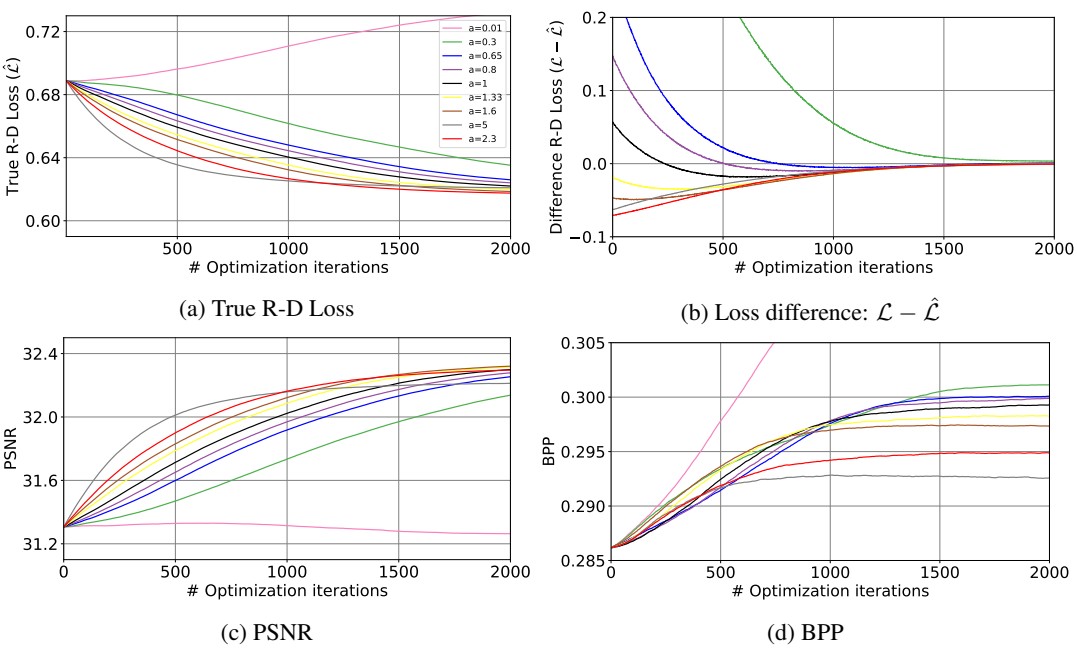

(a) True R-D Loss

(b) Loss difference: $\mathcal{L} - \hat{\mathcal{L}}$

(c) PSNR

(d) BPP

Figure B.4: Interpolation performance plots of different $a$ settings for SLL (a) True R-D Loss (b) Difference in loss (c) PSNR (d) BPP.

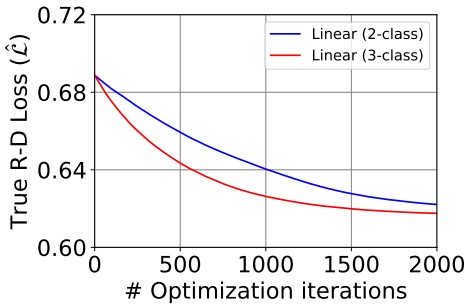

Figure B.5: True R-D loss curves for two- versus three-class rounding of the linear method.

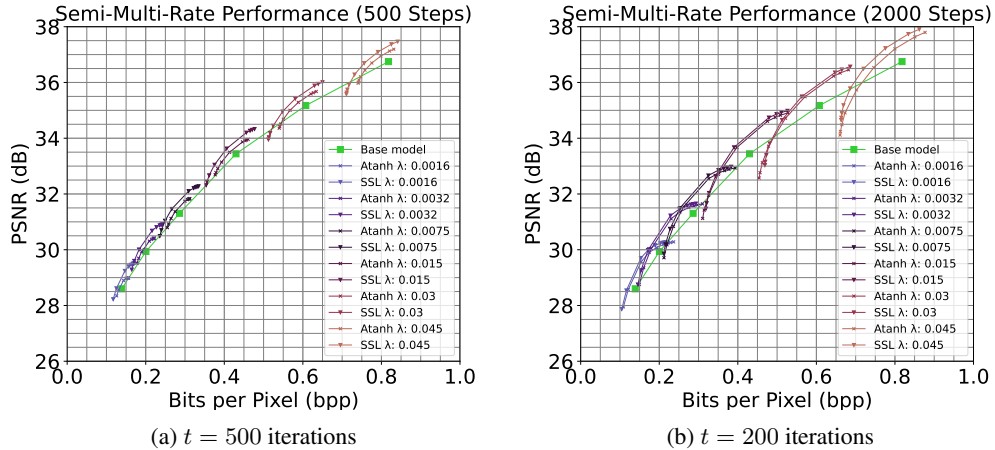

(a) $t = 500$ iterations

(b) $t = 200$ iterations

Figure B.6: R-D performance on Kodak of the Cheng et al. (2020) model when varying the target $\lambda$. Best viewed electronically.

# C   Mean-Scale Hyperprior

To make a clear comparison we trained a similar mean-scale hyperprior as in Yang et al. (2020). Therefore, we use the architecture of Minnen et al. (2018), except for the autoregressive part as a context model. Instead, we use the regular convolutional architecture of Ballé et al. (2018). The model package for the mean-scale hyperprior is from CompressAI Bégaint et al. (2020). The details and results of this model can be found in this section.

Similar as for Cheng et al. (2020) we run all experiments with temperature schedule $\tau(t) = \min(\exp\{-ct\}, \tau_{max})$. Additionally, we refine the latents for $t = 2000$ train iterations, unless specified otherwise.

**Implementations details**   The pre-trained mean-scale hyperpriors are trained from scratch on the full-size CLIC 2020 Mobile dataset Toderici et al. (2020), mixed with the ImageNet 2012 dataset Russakovsky et al. (2015) with randomly cropped image patches taken of size $256 \times 256$. For ImageNet, only images with a size larger than 256 for height and width are used to prevent bilinear up-sampling that negatively affects the model performance. During training, each model is evaluated on the Kodak dataset Kodak. The models were trained with $\lambda = \{0.001, 0.0025, 0.005, 0.01, 0.02, 0.04, 0.08\}$, with a batch size of 32 and Adam optimizer with a learning rate set to $1e^{-4}$. The models are trained for 2M steps, except for model $\lambda = 0.001$, which is trained for 1M steps and

Table C.1: True R-D loss results for the interpolation between different functions by changing $a$ of the SSL.

| a | R-D Loss |
|---|---|
| 0.01 | 1.15 |
| 0.3 | 0.7528 |
| 0.65 (approx atanh) | 0.7410 |
| 0.8 | 0.7396 |
| 1 (linear) | 0.7386 |
| 1.33 | 0.7380 ↓ |
| 1.6 (approx cosine) | 0.7382 |
| 2.25 | 0.7388 |
| 5 | 0.7415 |

model $\lambda = 0.08$, which is trained for 3M steps. Training runs took half a week for the 1M step model, around a week for the 2M step models, and around 1.5 weeks for the larger 3M step model. We ran all models and methods on a single NVIDIA A100 GPU. Further, the models for $\lambda = \{0.04, 0.08\}$ are trained with 256 hidden channels and the model for $\lambda = 0.001$ is trained with 128 hidden channels. The remaining models are trained with hidden channels set to 192.

## C.1   R-D Performance

We evaluate our best-performing method SSL on the Kodak and Tecnick datasets, by computing the R-D performance, average over each of the datasets. The R-D curves use image quality metric PSNR versus BPP on the Kodak and Tecnick dataset. Recall that as base model we use the pre-trained mean-scale hyperprior, trained with $\lambda = \{0.001, 0.0025, 0.005, 0.01, 0.02, 0.04, 0.08\}$. For SSL we choose $a = \frac{4}{3}$ as we found that this setting achieves the best R-D loss overall at 2000 iterations. This is a lower setting compared to the model by Cheng et al. (2020). The hyperparameters are similar to what we reported for Cheng et al. (2020) but with two main differences. We found that we could increase the learning rate by a factor 10 to 0.005 for atanh and SGA+. We also found that a lower $a \in [1.3, 1.4]$ was optimal.

**Kodak**   Figure C.2 shows the R-D curve for refining the latents, evaluated on Kodak. We compare SLL against baselines: STE, uniform noise, atanh and the base model at iteration $t = 500$ (see Figure C.2a) and after full convergence at $t = 2000$ (see Figure C.2b). As can be seen in Figure C.2a, STE performs slightly better than the base model, while after $t = 2000$ iterations the method performs worse, this is also reflected in the corresponding true loss curve for $\lambda = 0.01$ (see Figure C.1a), which diverges. Remarkably, for the smallest $\lambda = 0.001$, STE performs better than at $t = 500$. Adding uniform noise results in better performance when running the method longer. Comparing the R-D curves, Figure B.1 to Figure C.2, we find that most impact is made at $t = 500$ iterations. However, for the model similar to Yang et al. (2020) the performance lay closer to the performance of atanh, than for the model of Cheng et al. (2020).

**Tecnick**   Figure C.3 shows the R-D curve when refining latents on the Tecnick dataset, after $t = 500$ and $t = 2000$ iterations. As can be seen in the plot, we find that the longer the methods run, the closer the performance lies to each other. The improvement by SSL compared to atanh is greater for Tecnick than for Kodak.

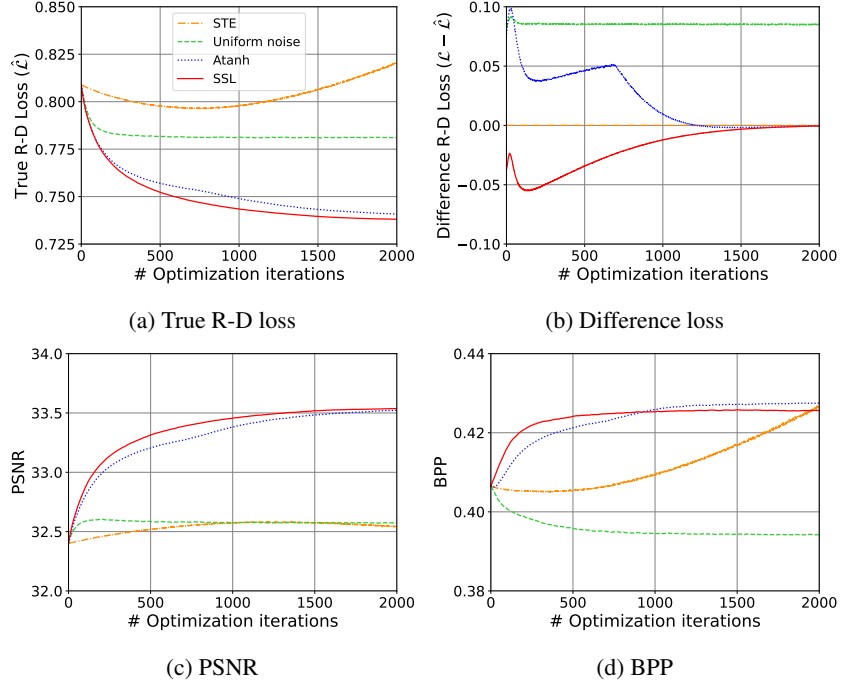

Figure C.1: Performance plots of (a) True R-D Loss (b) Difference in loss (c) PSNR (d) BPP.

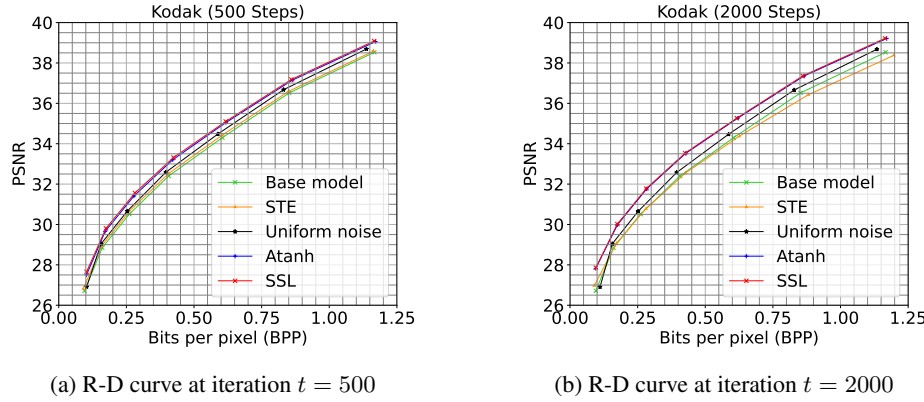

(a) R-D curve at iteration $t = 500$    (b) R-D curve at iteration $t = 2000$

Figure C.2: R-D performance on Kodak of the base mean-scale hyperprior model, STE, Uniform noise, SGA atanh and SSL with $a = \frac{4}{3}$.

**CLIC**    Figure C.4 shows the R-D curve when refining latents on the CLIC dataset, after $t = 500$ and $t = 2000$ iterations. Similar to the previous results, we find that the longer the methods run, the closer the performance lies to each other and that running the method shorter already gives better performance, compared to the base model.

**BD-Rate Gain**    In Table C.2, we computed the change in BD-PSNR and BD-rate for the mean-scale hyperprior model. We observe that SSL is slightly better than atanh, although the difference between them is smaller than reported on the Cheng et al. (2020) model. The gap between 500 steps and 2000 steps is also smaller compared to the results in 1.

## C.2    Analysis

In this appendix, we analyze additional experiments for the model, similar to those in Yang et al. (2020). The results for the analysis are obtained from a pre-trained model trained with $\lambda = 0.01$.

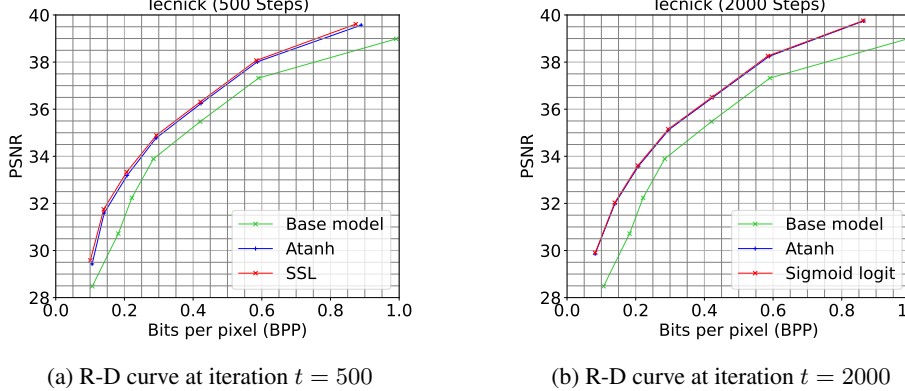

(a) R-D curve at iteration $t = 500$

(b) R-D curve at iteration $t = 2000$

Figure C.3: R-D performance on Tecnick of the base mean-scale hyperprior model, SGA $\mathrm{atanh}$ and SSL with $a = \frac{4}{3}$. Best viewed electronically.

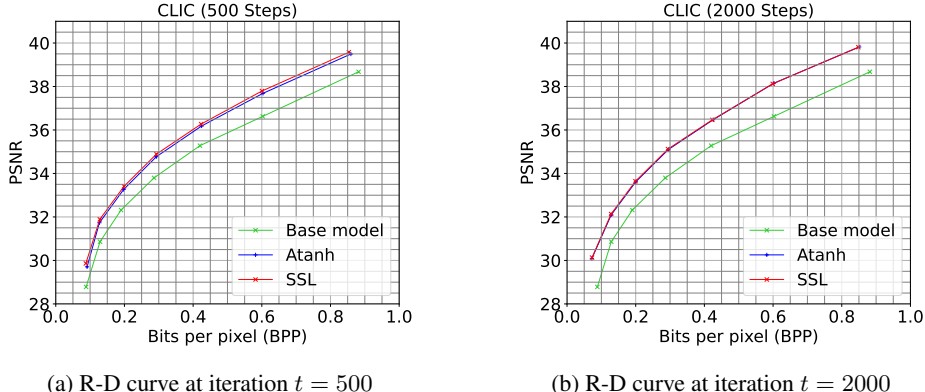

(a) R-D curve at iteration $t = 500$

(b) R-D curve at iteration $t = 2000$

Figure C.4: R-D performance on CLIC of the base mean-scale hyperprior model, SGA $\mathrm{atanh}$ and SSL with $a = \frac{4}{3}$. Best viewed electronically.

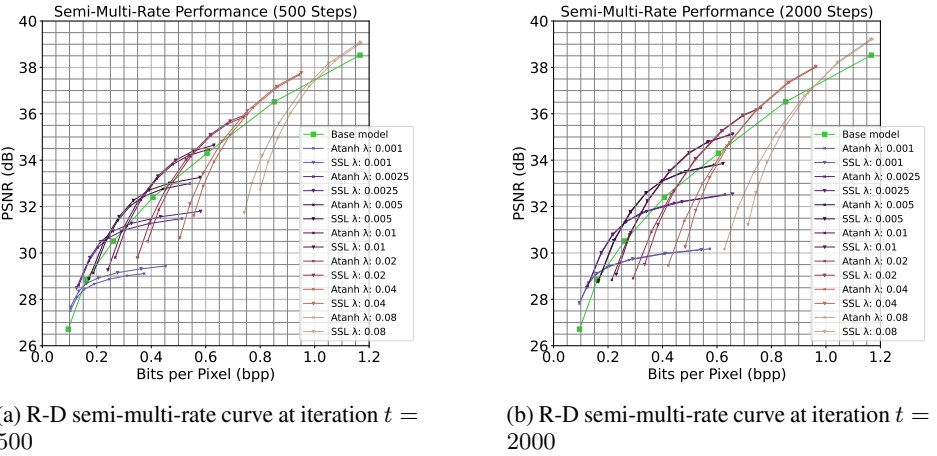

(a) R-D semi-multi-rate curve at iteration $t = 500$

(b) R-D semi-multi-rate curve at iteration $t = 2000$

Figure C.5: R-D performance on Kodak of the base mean-scale hyperprior model, SGA $\mathrm{atanh}$ and SSL with $a = \frac{4}{3}$. Each point is optimized with a different target $\lambda \in \{0.001, 0.0025, 0.005, 0.01, 0.02, 0.04, 0.08\}$.

Table C.2: Pairwise Comparison of BD-PSNR and BD-Rate for the Kodak, Tecnick, and CLIC dataset on the Mean-Scale Hyperprior Model.

| | BD-PSNR (dB) | | | | | | BD-Rate (%) | | | | | |
| | 500 steps | | | 2000 steps | | | 500 steps | | | 2000 steps | | |
| | Kodak | Tecnick | CLIC | Kodak | Tecnick | CLIC | Kodak | Tecnick | CLIC | Kodak | Tecnick | CLIC |
|---|---|---|---|---|---|---|---|---|---|---|---|---|
| Base vs SSL | 0.68 | 1.21 | 1.03 | 0.91 | 1.50 | 1.33 | -13.52 | -22.77 | -21.87 | -17.69 | -28.17 | -27.86 |
| Base vs Atanh | 0.59 | 1.06 | 0.89 | 0.87 | 1.46 | 1.30 | -11.90 | -20.20 | -19.14 | -17.03 | -27.49 | -27.28 |
| Atanh vs SSL | 0.09 | 0.15 | 0.14 | 0.04 | 0.04 | 0.03 | -1.80 | -3.22 | -3.14 | -0.82 | -1.03 | -0.74 |

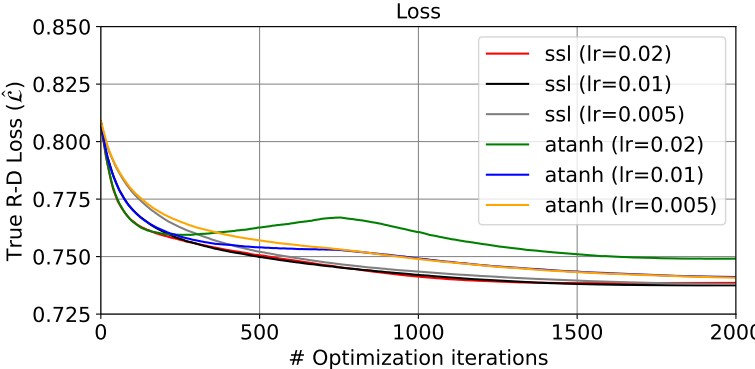

Figure C.6: True R-D loss curves for different learning rates settings for method SSL and atanh.

**Learning rates**  We run SSL and atanh with higher learning rate settings of 0.02 and 0.01 and compare it to the results obtained with learning rate 0.005. Figure C.6 shows the corresponding loss curves and Table C.3 shows the corresponding loss values at $t = \{500, 2000\}$ iterations. We find

Table C.3: True R-D loss for different learning settings of: atanh and SSL with $a = \frac{4}{3}$. At $t = 2000$ iterations and in brackets $t = 500$ iterations

| Lr \Method | SSL | atanh |
|---|---|---|
| 0.02 | 0.7386 (0.7506) | 0.7491 (0.7627) |
| 0.01 | 0.7375 (0.7498) | 0.7411 (0.7540) |
| 0.005 (base) | 0.7380 (0.7521) | 0.7408 (0.7570) |

that for a learning rate of 0.01 the gap between atanh versus SSL at 500 iterations is only around 14.3% smaller and it remains pronounced, while with a learning rate of 0.01 the gap at 2000 iterations is around 28.6% better. This concludes that SSL benefits more than atanh at 2000 iterations, with a higher learning rate. More interestingly, for a learning rate of 0.02, atanh diverges whereas SSL still reaches comparable performance. This highlights the sensitivity of atanh.

**Temperature sensitivity**  Table C.4 represents the stability of atanh and the SGA+ methods, expressed in true R-D loss, for different $\tau_{max}$ settings for the temperature schedule. As can be seen, the most optimal setting is with $\tau_{max} = 1$ for each of the SGA+ methods. atanh obtains equal loss for $\tau_{max} \in [0.4, 0.5]$. In general, we find that the linear method of SGA+ is least sensitive to changes in $\tau_{max}$ and has equal loss between $\tau_{max} \in$

Table C.4: True R-D loss for different $\tau_{max}$ settings of: atanh($v$), linear, cosine and SSL with $a = \frac{4}{3}$. Lowest R-D loss per column is marked with: ↓. Note that the function containing atanh is unnormalized.

| Function \ $\tau_{max}$ | 0.2 | 0.4 | 0.6 | 0.8 | 1.0 |
|---|---|---|---|---|---|
| $\exp \mathrm{atanh}(v)$ | 0.7445 ↓ | 0.7408 | 0.7412 | 0.7416 | 0.7418 |
| $1 - v$ (linear) | 0.7458 | 0.7406 ↓ | 0.7390 ↓ | 0.7386 | 0.7386 |
| $\cos^2(\frac{v\pi}{2})$ | 0.7496 | 0.7414 | 0.7393 | 0.7387 | 0.7384 |
| $\sigma(-a\sigma^{-1}(v))$ | 0.7578 | 0.7409 | 0.7391 | 0.7383 ↓ | **0.7380 ↓** |
| $\exp \mathrm{atanh}(v)$ | 0 | 0.0002 | 0.0022 | 0.0033 | 0.0038 |
| $1 - v$ (linear) | 0.0013 | 0 | 0 | 0.0003 | 0.0006 |

$[0.7, 1]$. To further examine the stability of the linear function compared to atanh, we subtract the best $\tau_{max}$, column-wise, from the linear and atanh of that column. We now see that the linear function is not only least sensitive to changes in $\tau_{max}$, but overall varies little compared to the best $\tau_{max}$ settings of the other methods. While the SSL has the largest drop in performance when reducing $\tau_{max}$, it achieves the highest performance overall for higher values of $\tau_{max}$.

**Interpolation**  Table C.1 presents the true R-D loss results for the interpolation with different $a$ settings for SSL for the mean-scale hyperprior model. In Figure C.7, the corresponding overall performance of the methods can be found. As can be seen in Figure C.7a, for $a = \{0.01, 0.30\}$, the functions diverge, resulting in large loss values. For $a = 0.65$, we find that the loss curve is slightly unstable at the beginning of training, which can be seen in the bending of the curve, indicating non-optimal settings. This may be due to the fact that we run all methods with the same $\tau_{max} = 1$ for a fair comparison. Additionally, note that SSL with $a = 0.65$ obtains a true R-D loss of $0.7410$ compared to $0.7418$ for $\mathtt{atanh}$ with the same settings. This is due to the fact that SSL, especially in the tails of the probability, is slightly more straight-curved compared to the $\mathtt{atanh}$ when looking at its probability space.

Remarkably, for $a \geq 1$, the difference in losses start close to zero (see Figure C.7b). SSL with $a = 5$ results in the fastest convergence and quickly finds a stable point but ends at a higher loss than most methods.

**Three-class rounding**  Table C.5 shows the true R-D loss for two- versus three-class rounding, at iteration $t = 500$ and in brackets $t = 2000$ iterations. For each method, we performed a grid search over the hyperparameters $r$ and $n$.

Table C.5: True R-D loss of two versus three-class rounding for SGA+ with the extended version of the linear, cosine, and SSL method at iteration 500 and in brackets after 2000 iterations.

| Function \Rounding | Two | Three |
|---|---|---|
| $f_{3c,\text{linear}}(w\|r = 0.98, n = 1.5)$ | $0.7552$ (0.7386) | $0.7517$ (0.7380) |
| $f_{3c,\text{cosine}}(w\|r = 0.98, n = 2)$ | $0.7512$ (0.7384) | $0.7513$ (0.7379) |
| $f_{3c,\text{sigmoidlogit}}(w\|r = 0.93, n = 1.5)$ | $0.7524$ (0.7380) | $0.7504$ (0.7380) |

Additionally, for the extended SSL, we also performed a grid search over $a$ and found the best setting to be $a = 1.4$. As can be seen in the table, most impact is made with the extended version of the linear of SGA+, in terms of the difference between the two versus three-class rounding at iteration $t = 500$ with loss difference $0.0035$ and $t = 2000$ with $0.0006$ difference. There is a small difference at $t = 500$ for the extended cosine version. In general, we find that running models longer results in convergence to similar values. SSL converges to equal values for two- and three-class rounding.

**Semi-multi-rate behavior**  Similar as in Appendix B.4, we experimented with different values for $\lambda$ to obtain a semi-multi-rate curve. For every pre-trained model, we ran SSL and $\mathtt{atanh}$ using $\lambda \in \{0.001, 0.0025, 0.005, 0.01, 0.02, 0.04, 0.08\}$. In Figure C.5, we have plotted the R-D curve of the base model (lime green line) and its corresponding R-D curves, obtained when refining the latents with the proposed $\lambda$'s for $\mathtt{atanh}$ and SSL. As can be seen, running the methods for $t = 500$ iterations, SSL obtains best performance. While the longer you train, the closer together the performance will be.

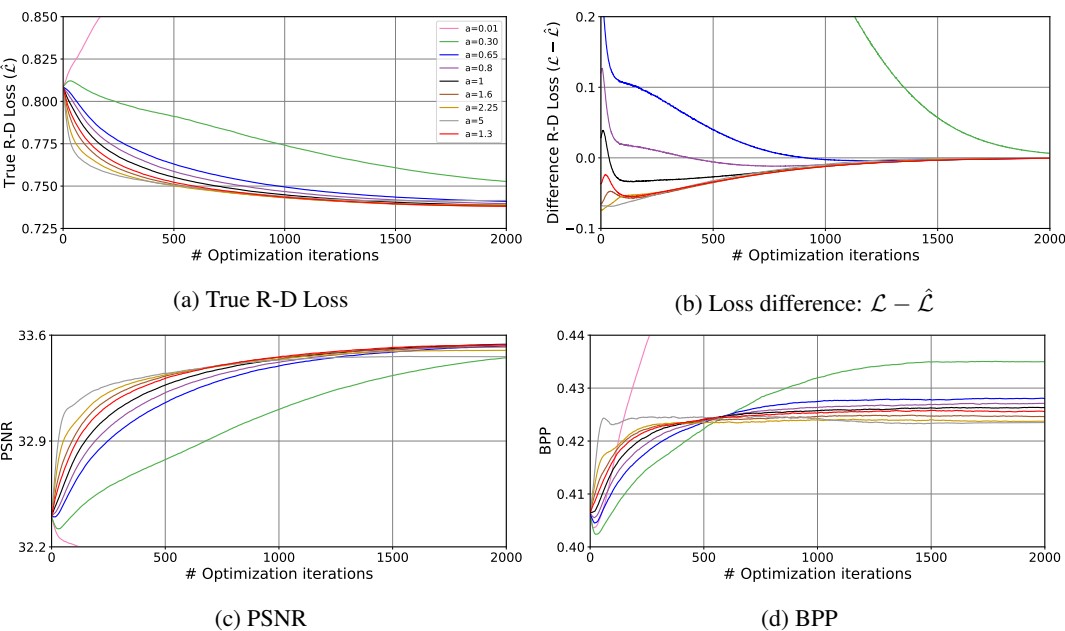

Figure C.7: Interpolation performance plots of different $a$ settings for SLL (a) True R-D Loss (b) Difference in loss (c) PSNR (d) BPP under the mean-scale hyperprior model.

