# OpenReview forum: "Robustly overfitting latents for flexible neural image compression"
_NeurIPS.cc/2024/Conference — NeurIPS 2024 poster_

### Official Review · Reviewer_Vgvs · 2024-06-29

**Soundness:** 3
**Presentation:** 2
**Contribution:** 2
**Rating:** 5
**Confidence:** 4

**Summary:**

The authors build on previous autoencoder-based neural image compression methods. Earlier works have shown that the quantized latent representations output by an end-to-end trained encoder for any given image are suboptimal for a decoder. Concretely, for a fixed set of decoder weights, one can usually find latent representations that yield a better rate-distortion than the representations output by the encoder. Hence, previous works have proposed optimizing the rate-distortion loss of an image using gradient descent for a fixed set of decoder weights. Since the latent representations we wish to optimize are discrete, previous works have developed "soft-quantization" approaches based on the Gumbel-Softmax trick, which introduces a temperature parameter $t$, such that for $t > 0$ the optimization problem becomes continuous, while $t \to 0$ recovers hard quantization. One starts with a temperature $t \gg 0$, which is then annealed towards $0$ during the optimization.

Specifically, the authors build on stochastic Gumbel annealing (SGA, Yang et al., 2020), which uses the function $f_\tau(x) = -\tanh^{-1}(x) / \tau$ to compute the soft quantization log-probabilities for a given temperature $\tau$.

In the present paper, the authors propose three alternatives for $f_\tau$ and conduct experiments analogous to the ones in Yang et al. (2020).

## References

Yang, Y., Bamler, R., & Mandt, S. (2020). Improving inference for neural image compression. Advances in Neural Information Processing Systems, 33, 573-584.

**Strengths:**

Unfortunately, I couldn't find any particular strengths worth highlighting.

**Weaknesses:**

Unless I have completely misunderstood the paper, the authors' motivation for the paper is incorrect. Concretely, for the function $\exp(-\tanh^{-1}(x))$ used by SGA, the authors claim on Line 155 that "The problem is that the gradients tend to infinity when the function approaches the limits of $0$ and $1$." While this is true for the limit point $1$, this does not hold for $0$. This is a crucial point, as the derivative tending to $\infty$ at $1$ is desirable, as it prevents rounding to the wrong value, while if this occurred $0$, it would cause optimization difficulties. However, it can be easily verified that the function I mentioned above doesn't have an infinite derivative at $0$ by plotting it. In fact, I am not sure what function the authors have plotted in Figure 1a under the name "atanh", as the graph of $\tanh^{-1}$ is not centrally symmetric.

In fact, the training instability of SGA arises because the derivative at $0$ becomes infinite as the temperature $\tau$ is annealed to $0$. All of the authors' proposed alternatives have the same issue, meaning they do not tackle the very problem they set out to solve.

The experiments are also not insightful and do not demonstrate that any of the proposed modifications lead to a significant improvement.

Furthermore, the paper's writing should be significantly improved. The authors review many unnecessary, basic details (such as the first two paragraphs of the introduction or the societal impact section, which should be cut altogether). The notation should also be cleared up; for example, the authors introduce duplicate or equivalent notations for the same objects: $\pi$ in Eq (3), $p(y)$ in Eq (4) for the soft quantization probabilities, $v_L$ and $w$ for the fractional part of $v$, and $n$ for the inverse temperature $\tau$. Finally, the font sizes should be increased in each figure as they are currently unreadable without zooming in significantly.

**Questions:**

n/a

---

> ### Author Rebuttal · Authors · 2024-08-04
>
> Thank you reviewer Vgvs for your comments.
>
> We think there is a misunderstanding, the reviewer believes that the probability function used in [1] only contains infinite gradients at 1. However, the reviewer may have overlooked the **normalization** from the Gumbel softmax that causes infinite gradients at 0. See the proof below, this will be included in the appendix.
>
> Recall that the probability given by a 2-class softmax is defined as:
> $$
> K(v) = \frac{e^{f(v)}}{e^{f(v)} + e^{g(v)}},
> $$
> where $f(v) = -\text{atanh}(v)$ and $g(v)= -\text{atanh}(1-v)$. We will study the softmax for the first class $0$, since the softmax is symmetric this also holds for the second class. The problem is that the gradients of the function $K(v)$ will tend to $-\infty$ for both $v \to 1$ (as agreed by the reviewer) but also for $v \to 0$. Here we show that the gradients also diverge for $v \to 0$, via the normalization with the term $g(v)$. First, take the derivative to $v$:
> $$
> \frac{d K(v)}{dv} = \frac{dK(v)}{df(v)} \cdot\frac{df(v)}{dv} + \frac{dK(v)}{dg(v)} \cdot\frac{dg(v)}{dv},
> $$
> where $\frac{dK(v)}{df(v)} = K(v)(1-K(v))$ and $\frac{dK(v)}{dg(v)} = -K(v)\frac{e^{g(v)}}{e^{f(v)} + e^{g(v)}}$. Recall that $\frac{d\text{atanh}(v)}{dv} = \frac{1}{1 - v^2}$ therefore $\frac{df(v)}{dv} = - \frac{1}{1-v^2}$ and $\frac{dg(v)}{dv} = \frac{1}{1-(1-v)^2 }$.
> Plugging this in and computing $\frac{dK(v)}{dg(v)}$ gives us:
> $$
> \frac{dK(v)}{dv}= K(v)(1-K(v))\left(- \frac{1}{1-v^2}\right) -K(v) \cdot \frac{e^{g(v)}}{e^{f(v)} + e^{g(v)}} \cdot \frac{1}{1-(1-v)^2 }.
> $$
> Taking the limit to $0$, (recall that $\lim_{v \to 0} K(v) = 1, \lim_{v \to 0} e^{f(v)} = 1$ and $\lim_{v \to 0} e^{g(v)} = 0$) allows the following simplifications:
> $$
> \lim_{v \to 0} 0 \cdot \left(-\frac{1}{1-0^2} \right) - 1 \cdot \frac{e^{g(v)}}{1 + 0} \cdot \frac{1}{1-(1-v)^2 }
> $$
> For simplicity we substitute $q=1-v$ (when $q \to 1$, then $v \to 0$) which will result in the following:
> $$
> \lim_{v \to 0} -e^{-\text{atanh}(1 - v)} \cdot \frac{1}{1-(1 - v)^2} = \lim_{q \to 1} -e^{-\text{atanh}(q)} \cdot \frac{1}{1-q^2 },
> $$
> Recall, $-\text{atanh}(q) = -\frac{1}{2}\ln{\frac{1+q}{1-q}}$, so $e^{-\text{atanh}(q)} = 1 / \sqrt{\frac{1 + q}{1 - q}}$ thus:
> $$
> -\lim_{q \to 1} \sqrt{\frac{1 - q}{1 + 1}} \cdot \frac{1}{1-q^2 } = -\lim_{q \to 1} \sqrt{\frac{1}{2}\frac{(1 - q)}{(1-q^2)^2} }
> $$
> Since $\frac{1}{2}$ is a constant and $\lim_{x \to \infty} \sqrt{x} = \infty$ the final step is to simplify and solve:
> $$
> -\lim_{q \to 1} \sqrt{\frac{(1 - q)}{(1-q^2)^2}} = -\lim_{q \to 1}\sqrt{\frac{-1}{(q-1)(q+1)^2}} = -\infty.
> $$
> This concludes the proof that the gradients tend to $-\infty$ for $v \to 0$.
>
> Further, the reviewer mentions that "I am not sure what function the authors have plotted in Figure 1a under the name 'atanh' ". Again, this is after normalization from the softmax.
>
> Regarding the temperature of 0 in the Gumbel, this is a separate issue and the temperature is therefore annealed to values above 0, both in [1] and our paper.
>
> Regarding the experiments, if the reviewer has more suggestions, we are happy to include them. The experiments do show issues with the sensitivity of [1].
>
> Regarding notations: as we mention in the paper, directly above Eq. (3):  $\log \pi$ represents unnormalized log probabilities. In contrast $p(y)$ is normalized. We will define Eq. (3) with normalization and with $p(y)$ for clarity. Regarding $v_L, v_R, w$, these are different $v_L, v_R \in [0, 1]$ whereas $w \in [-0.5, 0.5]$. $w$ is more helpful to define the 3-class because it has a center class. It is fair to point out that $n$ and $
> \tau$ can be fused into a new temperature in this function and we will point this out. However, following [1] ([https://github.com/mandt-lab/improving-inference-for-neural-image-compression/blob/6a97aba5b17c70847465f5865bd9e2bf58ccbe73/sga.py](https://github.com/mandt-lab/improving-inference-for-neural-image-compression/blob/6a97aba5b17c70847465f5865bd9e2bf58ccbe73/sga.py) #L118) the temperature is used twice, once on the logits and once again in the Gumbel softmax. However, $n$ is only applied to logits to scale the function and not to modify the Gumbel temperature.
>
> [1] Yang, Y., Bamler, R., \& Mandt, S. (2020). Improving inference for neural image compression. Advances in Neural Information Processing Systems, 33, 573-584.

---

> > ### Comment · Reviewer_Vgvs · 2024-08-10
> >
> > I thank the authors for their rebuttal. Their response helped me realise that I misunderstood not only their paper but also the original SGA paper (I was familiar with it but never implemented it myself). I never realised that the original SGA paper had such a glaring suboptimality. As I now understand, the `atanh` reparameterisation proposed by the authors of the SGA paper is superfluous, as the authors did not realise that $v - \lfloor v \rfloor$ is already in the correct range. Could the authors comment on their understanding of why the `atanh` reparameterisation could have been chosen?
> >
> > After the authors' rebuttal, I also re-read the paper and now believe that they make a valuable contribution. I have updated my score accordingly.
> >
> > My main concerns that I would like to see the authors address in the camera-ready version of the papers are:
> >  - To avoid colossal misunderstandings like mine above, I would like to ask the authors to update the manuscript and be crystal clear about what function produces the probabilities and what is being plotted in Fig 1a.
> >  - Improve the Figures by making them vector instead of raster graphics and improving their readability by increasing their tick and legend font sizes, and increasing the line widths.

---

> > > ### Author Response · Authors · 2024-08-12
> > >
> > > Thank you reviewer Vgvs for your reply and for updating your score.
> > >
> > > To answer your question regarding why the $\text{atanh}$ reparametrisation could have been chosen. This may be due to the fact that in the unnormalized log space, the $\text{atanh}$ looks like an appropriate function that satisfies some useful, and non-trivial properties, as mentioned in their paper. For example, the function $-\text{atanh}(v)$ being strictly decreasing on $(0,1)$ guarantees that the closer some value $v$ gets to an integer, the higher the probability it gets rounded to that $v$.
> > >
> > > Thank you for pointing out the following points, we will adjust these to avoid misunderstanding in the future:
> > > - We will add a clear explanation on how the probabilities of the functions in Fig 1a) are created.
> > > - We will improve the readability of each figure accordingly.
> > >
> > > We hope this answers your question. Feel free to reach out for any further comments.

---

### Official Review · Reviewer_5M8z · 2024-07-12

**Soundness:** 3
**Presentation:** 4
**Contribution:** 3
**Rating:** 7
**Confidence:** 3

**Summary:**

This paper proposes a technique which takes a neural compression method based on a VAE and runs an optimization using the same loss function as the original method was trained with, but instead of back-propagating into the weights of the network, the gradients accumulate into the quantized latents. Due to the fact that the latents are quantized, the authors propose 3 alternatives on how to handle this non-trivial issue. The results presented show that the method is improving RD (as one would expect) over the original latents, but the computational cost per image is rather high.

**Strengths:**

- The paper presents some relatively straight-forward approximations which should be easy to implement in practice
- Strong results, and good ablations in the appendix (I actually had a bunch of comments I ended up deleting because the answers were in the appendix, so I think it's a very detailed paper from multiple respects)

**Weaknesses:**

- I am rather confused by the motivation behind 3-class rounding.
- I would have liked to see a discussion on whether it's necessary to apply this algorithm to the hyper latents or not (technically the hyper latents can be computed from the latents, but I am unclear whether they don't suffer from the same problem as the latents)
- The fact that there are 6 distinct possible methods that actually get presented is a bit confusing (2-class + 3-class) * 3 variants. The appendix does go into ablating over all of these, which is great, but it's still a bit confusing.
- The method is acknowledged to be very expensive, and the gains are modest (though the authors do mention that this is a generic method that can be applied to any base architecture).

**Questions:**

- What made you choose the 3 probability functions you chose? Technically there could be many functions $p$ which would fit the requirements.
- Given the ablations, I saw that the 3-class rounding offers a modest improvement over 2-class, while being quite a bit more confusing to understand. Are there any other advantages besides "better RD performance" of 3-class rounding (i.e., do you need fewer iterations before convergence, etc)?

**Limitations:**

The authors have addressed the limitations of their method adequately. The only comment here would be that the energy impact of using such a method on a global scale would be quite large, while the reduction in bits for the amount of energy used to achieve this would be... questionable, unless one were to chase the absolute limits of compression.

---

> ### Author Rebuttal · Authors · 2024-08-04
>
> Thank you reviewer 5M8z for your comments and questions. The reviewer thinks the method is straightforward and easy to implement.
>
> _To answer the weaknesses:_
>
> Concerning point 1: the motivation behind the 3-class rounding. Besides our scientific curiosity, we did find that 3-class rounding improved performance. When there is a limited budget, it may be worth it to use the 3-class, since at earlier iterations, this method converges a bit faster than its 2-class counterparts. We will add a loss plot in the paper that shows the difference between the 2- versus 3-class rounding. The added complexity may not be worth it, which is why we see the 3-class as an extended version of the main functions. Therefore, we always recommend the 2-class linear as a starting point that does not require much tuning, we will highlight this in the paper. We will include a loss plot over the iterations in the paper that shows the difference between the 2- versus 3-class rounding.
>
> Regarding point 2, whether it is necessary to apply this algorithm to the hyper latents: We performed a run using the linear 2-class approach where we optimized only the latents (and not the hyper latents). We found a loss of 0.6233 when optimizing only the latents, and a loss of 0.6220 when optimizing both latents and hyper latents. We will add a discussion of this in the paper.
>
> Regarding point 3: for the six distinct methods, we wanted to be precise and show the possible options and their possible equivalences (such as SSL being able to interpolate between the other functions, and 3-class having settings that match the 2-class). Nevertheless, we understand this can be overwhelming and we hope that the recommendation from the point above (for the 2-class linear as a starting point) helps readers to make a quick and robust choice.
>
> _To answer the questions:_
>
> Regarding point 1, the main reason behind the three probability functions: We tried various methods while looking into the probability space. The main reason behind the linear version is the fact that it is the only function with constant gradients which is also the most robust choice, the cosine version is approximately mirrored across the diagonal of the $-\text{atanh}(x)$ which shows that it is more stable compared to the $-\text{atanh}(x)$, and the reason behind the SSL is that it is a function that can interpolate between all possible functions and can be tuned to find the best possible performance when necessary. We will add this in the paper.
>
> Regarding point 2: additional advantages of 3-class rounding, see our answer for weaknesses 1. We will add a loss plot over iterations, where the 3-class approach indeed converges faster.
>
> We hope this clarifies your points. Feel free to reach out for any further comments.

---

### Official Review · Reviewer_N6DE · 2024-07-19

**Soundness:** 3
**Presentation:** 3
**Contribution:** 3
**Rating:** 6
**Confidence:** 4

**Summary:**

In this paper, the authors proposed the method to improve the compression performance of the pre-trained end to end neural image compression methods by fine-tuning the latents of each image at the test time with the rate-distortion loss. They proposed three class rounding method, named SGA+ which is an extension of stochastic Gumbel annealing (SGA). The authors demonstrated that their method on the two pre-trained models with two different datasets and showed that their method was able to achieve better R-D performance than the existing methods.

**Strengths:**

1. The proposed SGA+ method has the potential to round the latent's to slightly far way quantization grid. This might be optimal with respect to the rate-distortion loss function. The modeling of the probability for three classes are well formulated.
2. modeling of the probabilities using three methods to overcome the limitations of the gradients in the corners of the $atanh$ function
2. Interpolation based function using sigmoid scaled logit to model the probabilities

**Weaknesses:**

1. The authors described the existing SGA method uses $atanh$ function to model the probabilities and as a result the gradients to tend infinity at the borders, and they proposed different ways to counter this problem by different functions. The instability of the gradients occurs only when the discretization gap is very minimal which might be the case at the higher BPP points in the R-D curve, so in these regions proposed method should provide higher gain. It is not evident from the results whether the proposed are able to achieve this?

2. The authors did not provide the B-D rate gain of their method with respect to the existing methods to quantify the average compression gain.
3. Analysis of what is percentage of the latents that were assigned into each category of three classes is missing to see the advantage of the proposed method.
4. Results are missing with recent end to end neural image compression methods.

5. The authors missed few works in the related section
 [a] M. Balcilar et. al, "Latent-Shift: Gradient of Entropy Helps Neural Codecs", 2023 IEEE International Conference on Image Processing (ICIP).

**Questions:**

1. Whether the gradients problem of the SGA at the corners cannot be solved with tuning the temperature parameter?

2. The results of STE method is not consistent with iterations at 500 and 2000. At 500 iterations, the STE is above the base model and at 2000 iterations is below the base model. what is the reason for this behavior? Is the baselines are tunned properly? It seems there is a divergence in the optimization.

3.  The analysis of BD rate gain [b] is important to quantify the average gain of the proposed method.

4. what are the differences and similarities between the Trellis Coded Quantization (TCQ) and your proposed method? In TCQ, also you have the possibility to have rounding the more classes. There are few works which make TCQ differentiable in the deep learning framework [c]

[b] G. Bjontegaard, "Calculation of average PSNR differences between RD-curves", VCEG-M33, Austin, TX, USA, April 2001.
[c] Deep Learning-based Image Compression with Trellis Coded Quantization   https://arxiv.org/pdf/2001.09417

**Limitations:**

Yes, limitations are addressed

---

> ### Author Rebuttal · Authors · 2024-08-04
>
> Thank you reviewer N6DE for your comments and questions.
>
> _To answer the weaknesses:_
>
> Regarding 1, whether the method might work better for higher BPP points in the R-D curve: For certain experiments this seems to be the case (e.g. on Kodak), for other experiments not. Due to the complexity of optimization, it is very difficult to predict which type of model suffers more from these spikes. E.g. it could also be the case that a lower BPP model would be more sensitive to gradient issues.
>
> Regarding 2: Thank you for pointing out that we did not provide the B-D rate gain of our method with respect to the existing methods. We will include these results in the attached PDF file.
>
> Regarding 3: The reviewer suggests an analysis of what percentage of the latents were assigned to the 3-classes. Therefore, we ran an extra experiment for the best settings of the 3-class extended version of the linear with $r=0.98$ and $n=2.5$. At the first iteration, the probability is distributed as follows: $p(y=\lfloor v \rceil) = 0.9329$, for $p(y=\lfloor v \rceil -1)= 0.0312$, and  $p(y=\lfloor v \rceil +1)=0.0359$. This indicates that the class probabilities are approximately $3.12\%$ for class $-1$ and $3.6 \%$ for class $+1$. This is a lot when taking into account that many samples are taken for a large dimensional latent.
>
> Regarding 4: Missing results with recent end-to-end neural image compression methods. We choose two different models to show the effect of latent optimization. The first model is trained from scratch (see Appendix), which is similar to the one trained in [1] to make a fair comparison. The other model is a pre-trained model for which we also showed similar improvements. We acknowledge that it would be interesting to see the method on other models as well.
>
> Regarding 5: Missing a few works in the related section. Thank you for pointing this out, we will add [2] in the paper.
>
> _To answer the questions:_
>
> Regarding 1: the gradients problem at the corners, by tuning the temperature parameter. This is difficult. The problem is that when the temperature increases (which traditionally stabilizes) the diverging gradient of the atanh logits becomes worse. Lowering the temperature mitigates the gradient issue but makes the Gumbel sampling itself more unstable. Experimentally, we demonstrate this in Table 1, where atanh logits over a wide range of temperatures lead to worse performance.
>
> Regarding 2: the reason for the behavior of the STE method. We tuned the STE method, just as the other baselines. However, the STE method is the only method that has a lot of trouble converging. Even with smaller learning rates, the method performed poorly.
> The instability of training is not only observed by us, but is something that is also mentioned in [1] and [4]. In [1] they have tried to overcome this following [4], by changing the gradient of the backward pass to (clipped)ReLU instead of using the identity. However, this did not work. We will highlight this clarification in the paper.
>
> Regarding 3: The missing B-D rate gain, we will add this in the paper and included the results in the attached PDF file, as mentioned above.
>
> Regarding 4: Trellis-coded quantization (TCQ) is related to our approach in that regard. They also consider a softmax over different quantization levels, although they consider all possible quantization levels. However, while we maintain a scalar quantization approach, TCQ is a method to more efficiently perform vector quantization (VQ). Their rounding approach is also not stochastic, but deterministic based on the optimal Trellis path. We will add a discussion in the paper concerning [3].
>
> We hope this clarifies your questions. Feel free to reach out for further comments.
>
> [1] Yang, Y., et al., "Improving inference for neural image compression", 2020 Advances in Neural Information Processing Systems (NeurIPS)
>
> [2] M. Balcilar et. al, "Latent-Shift: Gradient of Entropy Helps Neural Codecs", 2023 IEEE International Conference on Image Processing (ICIP)
>
> [3] Deep Learning-based Image Compression with Trellis Coded Quantization [https://arxiv.org/pdf/2001.09417](https://arxiv.org/pdf/2001.09417)
>
> [4] Yin, P., et al., "Understanding Straight-Through Estimator in Training Activation Quantized Neural Nets", 2019 International Conference on Learning Representations (ICLR)

---

### Author Rebuttal · Authors · 2024-08-04

Contains an additional experiment requested by reviewer N6DE.

---

### Decision · Program_Chairs · 2024-09-25

**Decision:**

Accept (poster)

**Comment:**

This paper presents three improvements to stochastic Gumbel annealing, which is used to refine the discrete latents in neural compression methods. Reviewers appreciated the simplicity of the method, solid improvements and extensive ablations. Initially there was some misunderstanding on the part of some reviewers that was addressed in the discussion, and graciously acknowledged by the reviewers. Overall this paper represents a solid contribution to the neural compression literature.